

# Seasonal variation of eddy activity and associated heat/salt transport in the Bay of Bengal based on satellite, Argo and 3D reprocessed data

Wei Cui[1], Jie Zhang[1], and Jungang Yang [1*]

[1] First Institute of Oceanography, Ministry of Natural Resources, Qingdao 266061, China

*Correspondence to*: Jungang Yang (yangjg@fio.org.cn)

**Abstract.** Based on satellite altimetry data spanning over 26 years in combination with Argo profile data or three-dimensional (3D) reprocessed thermohaline fields, the eddy synthesis method was used to construct vertical temperature and salinity structures of eddies in the Bay of Bengal, and the seasonal thermohaline properties of eddies and the heat and salt transport by eddies were analyzed. Analysis revealed that mesoscale eddy activity in the Bay of Bengal has evident seasonal
variation, which has notable impact on the circulation system within the entire bay. Temperature anomalies caused by eddies are usually between ±1°C and ±3°C, positive for anticycloic eddies (AEs) and negative for cyclonic eddies (CEs), and the magnitude varies seasonally. Salinity anomalies caused by eddies are small and disturbance signals in the southern bay due to the small vertical gradient of salinity there; salinity anomalies in the northern bay are generally between ±0.2 psu and ±0.3 psu, negative for AEs and positive for CEs. Owing to obvious seasonal changes of both the eddy activity and the vertical
thermohaline structure in the Bay of Bengal, the eddy-induced heat and salt transport in different seasons also changes substantially. Generally, high heat and salt transport is concentrated in eddy-rich regions, e.g., the western, northwestern and eastern parts of the bay, the seas to the east of Sri Lanka, and the region to the southeast outside of the bay. The southern part of the bay shows weak freshwater transport owing to the inconsistent salinity signal within eddies. The seasonal *ZHT* of CEs and AEs in the whole bay is in the order of $10 \times 10^{12}$ W, with higher values in autumn and winter and smaller values in spring
and summer. The seasonal *ZWT* of CEs is generally larger than that of AEs, thus the *ZWT* of all eddies is eastward with high values ($> 5 \times 10^3$ m$^3 \cdot$s$^{-1}$) in summer and autumn. By contrast, the seasonal *MHT* and *MWT* in the whole bay is relatively small, mostly below $1 \times 10^{12}$ W and $2 \times 10^3$ m$^3 \cdot$s$^{-1}$, respectively. This work provides data that could support further research on the heat and salt balance of the entire Bay of Bengal.

## 1 Introduction

Oceanic mesoscale eddies are rotating coherent structures of ocean currents, which generally refer to ocean features with spatial scales from tens to hundreds of kilometers and time scales from days to months (Robinson, 2010). Following recent advances in remote sensing satellites and the abundance of in situ observational data, it has been established that mesoscale eddies can be found nearly everywhere in the world's oceans (Chelton et al., 2011a; Xu et al., 2011; Fu, 2009; Chaigneau et al., 2009), and they transport water, heat, salt, and other tracer materials as they propagate in the ocean,
impacting water column properties and biological activities (Chelton et al., 2011b; Xu et al.,2011; Dong et al., 2014). Combinning altimetry data with Argo profile data, Zhang et al. (2014) found that mesoscale eddies have strong zonal mass transport which was comparable in magnitude to that of the large-scale wind- and thermohalinedriven circulation.



The Bay of Bengal is located at the northeastern part of the Indian Ocean. The northern Indian Ocean is subject to monsoonal wind forcing, which means there is a near complete reversal of winds from summer (the Southwest Monsoon) to winter (the Northeast Monsoon) and the ocean circulation responds accordingly. During the summer Southwest Monsoon, the upper ocean circulation from south of the equator to the northern boundary is eastward. The eastward flow at the southern India is called the Southwest Monsoon Current (SMC). During the winter Northeast Monsoon, a westward flow, the Northeast Monsoon Current (NMC), appears along the south side of Sri Lanka and India. Affected by complex exogenous effects such as the local monsoon, equatorial remote forcing and seasonal changes in river runoff, the circulation of the Bay of Bengal has obvious seasonal variation (Hacker et al., 1998; Eigenheer and Quadfasel, 2000; Somayajulu et al., 2003; Qiu et al., 2007; Cheng et al., 2013; Chen et al., 2017). The climatological monthly mean circulation structure and thermohaline properties of the Bay of Bengal are shown in Figure 1. During the summer Southwest Monsoon, alternate cyclonic and anticyclonic circulation cells prevail in the western bay; a basin-scale cyclone-like gyre dominates the bay during the November monsoon transition; during the winter Northeast Monsoon, the cyclonic gyre weakens, and an anticyclonic circulation appears in the northern bay; in spring premonsoon, the bay is again dominated by a strong anticyclonic gyre. The East Indian Coastal Current (EICC), i.e., the western boundary current in the Bay of Bengal, reverses direction twice a year, flowing northeastward in the Southwest Monsoon and southwestward in the Northeast Monsoon.

Many studies show that there are abundant mesoscale eddies associated with the seasonally circulations in the Bay of Bengal (Babu et al., 1991; Prasanna Kumar et al., 2004; Chen et al., 2012 & 2018; Cui et al., 2016; Cheng et al., 2018). Somayajulu et al. (2003) analyzed the seasonal and inter-annual variability of surface circulation in the Bay of Bengal, and found that the monsoon conversion, EICC instability, as well as the coastally trapped Kelvin waves and radiated Rossby waves are responsible for the the observed variability of the mesoscale eddies in the bay. Chen et al. (2018) suggested that both local monsoonal winds and remote equatorial winds, and ocean internal instability are the main reasons for the generation and modulation of eddy kinetic energy in this region. The upper seasonal circulation in the Bay of Bengal is driven by the monsoon, on which are superimposed by the local Ekman drifting and geostrophic circulation, so its seasonal changes are not completely synchronized with the monsoon transition (Vinayachandran et al., 1999; Qiu et al., 2007; Sreenivas et al., 2012). The characteristics of the Bay of Bengal circulation and thermohaline structure are important for understanding the mesoscale eddy activity in this area.

The sea surface temperature (SST) in the Bay of Bengal has obvious seasonal variation, showing a bimodal distribution, which is influenced by the inflows from the tropical Indian Ocean and Arabian Sea to the South, considerable river runoff to the North, and abundant precipitation (Graham and Barnett, 1987; Rao et al., 2002; Shenoi et al., 2002; Murty et al., 1998). The lowest SST throughtout the entire bay occurs in winter, when there is a slight increase from north to south. In spring, the SST rises throughout the entire bay reaching 29°C in May. During the summer Southwest Monsoon, the cold upwelled water along the coasts is transported by the Southwest Monsoon Current into the Bay of Bengal, the SST in the Bay of Bengal is not high in summer. As the Southwest Monsoon weakens in autumn, the SST increases apparently in the northern bay. The salinity in the Bay of Bengal decreases from about 34 psu at about 5°N to 30 psu or less in the north. The Bay of Bengal with the fresh waters is dominated by the considerable runoff from all of the major rivers of India, Bangladesh, and Burma (Varkey et al.,1996; Prasad, 1997; Rao et al., 2003). The seasonal barrier layer in the Bay of Bengal is brought about by the strong salinity stratification due to the influx of freshwater from river discharge and excess precipitation over evaporation



(Kumari et al., 2018; Vinayachandran et al., 2002; Akhil et al., 2014).

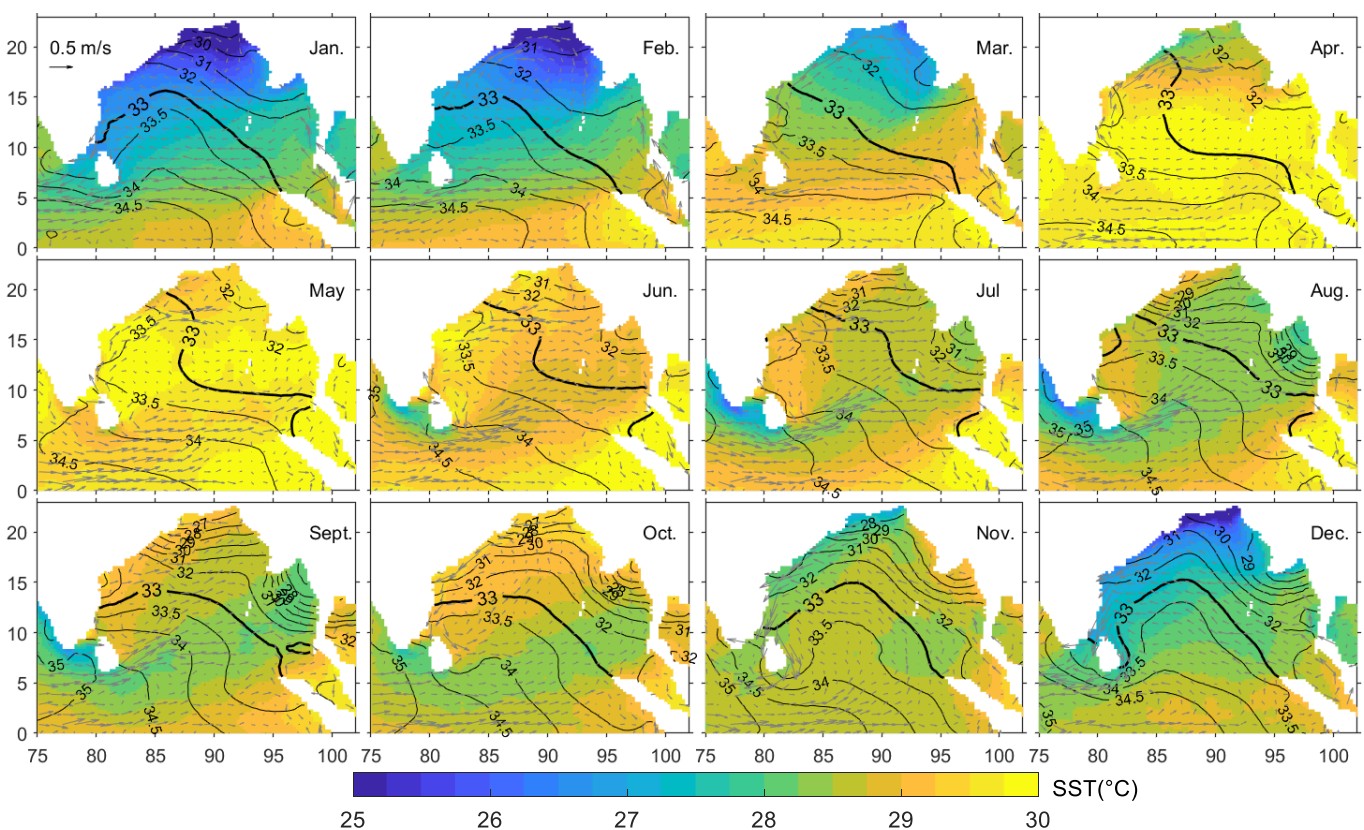

Figure 1: Climatological monthly sea surface temperature fields (color), mean surface currents (arrows), and surface salinity (contours) in
the Bay of Bengal. The climatological sea surface temperature fields are from monthly averaged OISST dataset with 0.25° regular grid at
global scale from Jan. 1982 to Dec. 2011 (Banzon et al., 2014; ftp://eclipse.ncdc.noaa.gov/pub/OI-daily-v2/). The climatological surface
currents are from monthly averaged global total velocity field (MULTIOBS_GLO_PHY_REP_015_004) at 0 m and 15 m with 0.25° grid
from Jan. 1993 to Dec. 2018 (Etienne, 2018). The climatological surface salinity fields are from the global SSS/SSD L4 Reprocessed
dataset (MULTIOBS_GLO_PHY_REP_015_002) with 0.25° grid from Jan. 1993 to Dec. 2018 (Mertz et al., 2018). The later two datasets
are available on http://marine.copernicus.eu.

The surface characteristics of oceanic eddies can be inferred from remote sensing data, and the vertical thermohaline
profile of subsurface waters can be provided by Argo buoys. In recent years, by combining satellite altimetry and Argo
profiling float data, analysis of the vertical structure of eddies has become an important part of the study of oceanic eddies
(Chaigneau et al., 2011; Liu et al., 2012; Yang et al., 2013; Amores et al., 2017). Knowledge of the vertical structure of the
ocean is vital both for comprehensive understanding of ocean dynamic processes and for analysis of the ocean circulation
and energy transport. Based on satellite altimetry and Argo floats, Lin et al. (2019) and Gulakaram et al. (2020) showed that
eddy-induced ocean anomalies in the Bay of Bengal are mainly confined to the upper 300 m and eddy thermohaline structure
has a seasonal character. Cui et al. (2021) found that the thermohaline properties of mesoscale eddies in the Bay of Bengal
are different in the north-south direction. Combining estimated eddy diffusivity from 25 years of altimetry data with
corresponding tracer gradients from the World Ocean Atlas 2013, Gonaduwage et al. (2019) investigated the meridional and



zonal eddy-induced heat and salt transport in the Bay of Bengal, and they found that the baroclinic instability, local wind-stress curl and remote forcing from the equator contribute to the seasonal modulation of eddy-induced heat transport.

Many studies examined the surface characteristics of eddies in the Bay of Bengal ans some have investigated the vertical eddy properties (Nuncio and Kumar, 2012; Dandapat and Chakraborty, 2016; Chen et al., 2012 & 2018; Cui et al., 2021). However, few studies considered the seasonal variation of the three-dimensional (3D) thermohaline structure and the heat and salt transport due to mesoscale eddies. Considering the hydrological differences from north of the bay to the south, the eddy vertical structure in different subregions should be further studied. Owing to the characteristics of the oceanic circulation and regional monsoons, there are obvious seasonal differences in eddy activity in the Bay of Bengal. Specifically, 
the seasonal variation of surface eddies, its 3D thermohaline structure, regional thermohaline structure variation, seasonal heat/salt transport and their spatial distribution characteristics have not been analyzed comprehensively.

    In this study, based on merged satellite altimetry data spanning over 26 years, the automatic identification method was used to extract information on the position and shape of mesoscale eddies in the Bay of Bengal, and the seasonal variation of the eddies was analyzed in detail. Then, by combining the satellite altimetry data with either Argo profile data or 3D 
thermohaline fields, the eddy synthesis method was used to construct the 3D thermohaline structures of eddies in the study area, their seasonal thermohaline properties and regional thermohaline variations were analyzed. Finally, based on eddy movement and thermohaline properties, the heat and salt transports by eddies were estimated, and their seasonal variation and spatial distribution characteristics were analyzed. The remainder of this paper is organized as follows. Section 2 describes the data and methods adopted in the study. Section 3 presents the seasonal variations and seasonal 3D thermohaline 
properties of the eddies. Section 4 analyzes the seasonal heat and salt transports by eddies in the Bay of Bengal. Finally, a summary and our conclusions are presented in Section 5.

## 2 Data and methods

### 2.1 Data

    The daily and monthly 0.25°×0.25° gridded sea level anomaly (SLA) product (SEALEVEL_GLO_PHY_L4_REP_ 
OBSERVATION_008_47) from January 1993 to February 2019 are used to determine the presence and positions of mesoscale eddies in the Bay of Bengal. The SLA product is processed by the Archiving, Validation, and Interpretation of Satellite Oceanographic data (AVISO) and distributed by the European Copernicus Marine Environment Monitoring Service (CMEMS, http://marine.copernicus.eu).

    The Argo float profiles provided by the Coriolis Global Data Acquisition Center of France (http://www.coriolis.eu.org) 
are used to analyze the vertical temperature and salinity structures of eddies. In the analysis, we have taken pressure, temperature, and salinity profiles with quality flag 1, and have followed Chaigneau et al. (2011) for the selection of the profiles from the year 2001 to 2019. The final dataset includes of total 29,219 available profiles in our study region. Potential temperature $\theta$ and salinity $S$ data in each profile were linearly interpolated onto 101 vertical levels from the surface to 1000 dbar with an interval of 10 dbar using the Akima spline method. To get the thermohaline structures of mesoscale eddies, 
potential temperature anomaly $\theta'$, and salinity anomaly $S'$ of Argo profiles were computed by removing Argo seasonal-mean climatologic profiles.

    The ocean reprocessed data can provide the 3D thermohaline information of the surface eddies captured by the satellite



altimetry. The Global ARMOR3D L4 Reprocessed dataset (MULTIOBS_GLO_PHY_REP_015_002, distributed by CMEMS, http://marine.copernicus.eu) consists of 3D temperature, salinity, heights and geostrophic currents, available on a 0.25° regular grid and on 33 depth levels from the surface down to the bottom (Guinehut et al., 2012). The ARMOR3D dataset is obtained by combining satellite (SLA, geostrophic surface currents, SST) and *in-situ* (temperature and salinity profiles) observations through statistical methods. The dataset is available as weekly means for the period 1993–2019. Similar to Argo profiles, the 3D temperature and salinity anomaly fields were computed by removing ARMOR3D seasonal-mean climatologic fields.

## 2.2 Eddy detection, 3D reconstruction, and heat-salt transport estimation

### 2.2.1 SLA-based eddy identification and tracking

In SLA fields, mesoscale eddies can generally be identified as regions enclosed by SLA contours. A geometric algorithm for eddy identification based on the outermost closed contour of an SLA has been proposed by Chelton et al. (2011a). Following the algorithm, an eddy is defined as a simply connected set of pixel grids that satisfying some criteria. For the Bay of Bengal, the minimum amplitude of an eddy is increased from the original 1 cm used by Chelton et al. (2011a) to 3 cm in this study. The reason for this change is that the accuracy of measuring heights using Jason series altimeters (including TOPEX/Poseidon and Jason-1/2/3), which currently have optimal performance for observing ocean dynamics, is only about 2 cm in the open sea (Dufau et al., 2016). Furthermore, the distance between the two furthest-apart internal points in an eddy is less than 600 km for avoiding enclose elongated regions.

Based on daily SLA fields, the mesoscale eddies in the Bay of Bengal are identified, and the eddy amplitude, eddy scale/radius, and eddy propagation velocity are quantified over the study area. The eddy amplitude is defined here to be the magnitude of the SLA difference between the eddy boundary and the eddy center (local extremum). The eddy scale/radius is defined as the equivalent radius of a circle with the same area which is delimited by the eddy boundary. Based on the eddy identification results in the continuous time series, the evolution process of eddies (eddy trajectories) in the ocean can be tracked by comparing the eddy positions and dynamic properties (Chaigneau et al., 2008; Henson and Thomas, 2008; Nencioli et al., 2010; Souza et al., 2011). For an eddy at day $n$, its trajectory is tracked by searching the most similar eddy at the subsequent day $n+1$ in terms of the type and eddy characteristics within a circle of eddy radius (Chaigneau et al., 2008; Cui et al., 2021). To avoid the false tracking of the eddies, the same eddy is searched continuously for 10 days with circles of growing radius (max double eddy radius in the $10^{th}$ day) when no match eddy is detected in subsequent time step $n+1$. The lifetime of an eddy represents the duration of an eddy from its generation to its termination. The eddy propagation velocity is defined as the change of the eddy center position as a function of time.

In addition, in order to study the stable eddies and their seasonal spatial distribution, monthly eddies are identified from the monthly SLA fields without trajectory tracking. For such monthly eddies, the tracking processing is not performed, and the monthly result identified from monthly SLA fields are processed as individual eddies.





Figure 2: (a) Reconstruction of 3D structure of an eddy based on sea level anomaly (SLA) fields and vertical temperature anomaly $\theta'$ and salinity anomaly $S'$ data. The top layer represents the SLA fields and identified eddies, where solid and dashed lines represent CEs and AEs, respectively. The colors in the lower layers represent the temperature anomaly $\theta'$ field at different depths from the ARMOR3D data. The vertical solid and dashed lines represent Argo profiles located in CEs and AEs, respectively, and each profile is marked by a letter in the top layer. (b, c) The graphs on the right show the vetical temperature and salinity anomaly profiles of the Argo buoy located in the eddies.



### 2.2.2  3D eddy reconstruction

The 3D structures of the eddies were constructed by surfacing the Argo float profiles into SLA-based eddy areas, as
shown in Figure 2. In this study, all eddies with a lifetime of ≥30 days and Argo profiles choosed following Chaigneau et al.
(2011) were used for eddy composition. We considered the detection results (from daily SLA fields) of the long-lived eddy to
match the Argo profiles on the same day, and selected Argo profiles with a distance of <1.5 radii from eddy center for
vertical eddy structure analysis. All the Argo profiles were classified into two categories according to eddy polarity.
Consequently, 3882 and 4097 Argo profiles were selected for cyclonic and anticyclonic eddy reconstruction, respectively.
For each Argo profile matched by an eddy, we calculated the relative zonal and meridional distances to the eddy center. The
relative distances were normalized relative to the eddy radius. Then, all the Argo profiles were transformed into the
normalized eddy coordinate space, and $\theta$, $S$ and $\theta'$, $S'$ data of Argo profiles were mapped onto 0.1×0.1 grid using inverse
distance weighting interpolation at each vertical level from the surface to 1000 dbar. Finally, composites of 3D thermohaline
structures were reconstructed in each normalized grid location.

Considering the hydrological differences from north of the bay to the south, here the Bay of Bengal is divided into north
and south subregions with 12°N as the boundary to study the eddy 3D structure of each subregion. In addition, weekly 3D
temperature and salinity field data from the ARMOR3D reprocessed dataset were also used to provide vertical structure
information on the surface eddies. We matched the eddy results identified from daily SLA fields with the weekly 3D field
data at the closest time such that we could obtain the 3D temperature and salt structure of each eddy (Figure 2a).

### 2.2.3 Eddy-induced heat and salt transport estimation

A nonlinear eddy can maintain its own water body characteristics and have minimal exchange with the surrounding
water mass as it propagates in an ocean. By combining the spatial-scale information of the eddies provided by the SLA fields
with the vertical temperature and salt anomaly information provided by the ARMOR3D temperature and salinity fields, the
heat anomaly $H_e$ and salt anomaly $S_e$ could be obtained for each eddy (the subscript $e$ means eddy):

$$H_e = \rho_0 C_{p0} \int_{-D_0}^{0} \iint_R \theta_e'(\Delta x, \Delta y, z) \, d(\Delta x) d(\Delta y) dz \tag{1}$$

$$S_e = \rho_0 \int_{-D_0}^{0} \iint_R s_e'(\Delta x, \Delta y, z) \, d(\Delta x) d(\Delta y) dz \tag{2}$$

Here, the mean upper ocean density and heat capacity are $\rho_0$ = 1025 kg·m⁻³, $C_{p0}$ = 4200 J·kg⁻¹·°C⁻¹. $\theta_e'$ and $s_e'$ are the
eddy-induced potential temperature anomaly and salinity anomaly, respectively. $R$ is the eddy region, $D_0$ is the intergration
depth (500 dbar for $H_e$ and 300 dbar for $S_e$; Lin et al. (2019); Gulakaram et al. (2020); also, Section 3.2). The unit of eddy
heat anomaly $H_e$ is J, and that of salt anomaly $S_e$ is psu·kg.

Instead of using eddy propagation velocity to calculate eddies' heat transport (Dong et al., 2014), eddy trajectories are
used to calculate transport by eddy movements (Dong et al., 2017). Here we use 0.25° grid cells to calculate the
eddy-induced heat and salt transport through following the eddy trajectory and check whether it crosses grid cell boundaries.
If an eddy crosses the western or eastern boundaries, it results in zonal transport; whereas eddy crossing of the northern or
southern boundaries results in meridional transport. In addition, the east and the north transport are defined as positive, while
the west and the south are negative. For a grid cell, the zonal heat and salt transport $Q_{hz}$ and $Q_{sz}$ (the subscript $z$ means zonal)
are equal to the sum of heat anomalies $H_e$ and salt anomalies $S_e$ of all eddies $i$ which cross the east or west boundary, divided
by the meridional length $D_m$ of the grid (the subscript $m$ means meridional; unit: m) and time length $T$ (unit: s) (Dong et al.,





2017):

$$Q_{hz} = \frac{\sum H_{ei}}{2D_m \cdot T} \tag{3}$$

$$Q_{sz} = \frac{\sum S_{ei}}{2D_m \cdot T} \tag{4}$$

Here the unit of eddy heat transport $Q_{hz}$ is W·m$^{-1}$, and that of eddy salt transport $Q_{sz}$ is psu·kg·m$^{-1}$·s$^{-1}$. The time length $T$ is 26 years corresponding to the time-series length of SLA products used for the eddy identification from Jan. 1993 to Feb. 2019. The denominator factor of 2 is because we separately considered the east and west boundaries of the eddy moving through the grid. Similarly, the eddy-induced meridional heat and salt transport $Q_{hm}$ and $Q_{sm}$ are calculated by

$$Q_{hm} = \frac{\sum H_{ei}}{2D_z \cdot T} \tag{5}$$

$$Q_{sm} = \frac{\sum S_{ei}}{2D_z \cdot T} \tag{6}$$

Here, $D_z$ is the zonal length of the grid.

In the actual calculation, a moving average filter with 1°×1° box size is applied to reduce noise. Salt transport can be treated as an equivalent freshwater volume transport assuming conservation of mass across the transport section (Dong et al., 2014), $F_w = -\frac{Q_s}{\rho_0 s_0}$, where mean upper ocean density and salinity are $\rho_0$ = 1025 kg·m$^{-3}$, $s_0$ = 35 psu, $Q_s = (Q_{sm}, Q_{sz})$ is a vector whose components are the meridional and zonal salt transports. The unit of $F_w$ is m$^2$·s$^{-1}$, which represents the freshwater volume flux per unit distance.

## 3 Seasonal variation of eddy activity in the Bay of Bengal

### 3.1 Seasonal spatial distribution of eddies

The Bay of Bengal is affected by the Southwest Monsoon and Northeast Monsoon, and its entire circulation system is characterized by monsoon circulation. Following many studies on the Bay of Bengal (Somayajulu et al., 2003; Patnaik et al., 2014; Seo et al., 2019), the seasons are defined as the Winter monsoon (December–February), Spring premonsoon (March–May), Summer monsoon (June–September), and Autumn postmonsoon (October–November) in the present study. Based on daily SLA fields spanning over 26 years (from January 1993 to February 2019), 620 cyclonic eddies (CEs) and 516 anticyclonic eddies (AEs) (eddy trajectories) with lifetimes ≥ 30 days in the Bay of Bengal were detected in the eddy tracking procedure. The seasonal distributions of eddy trajectories (Supplementary Material Figure S3) show that eddy activities have obvious seasonal variation, but messy trajectories obscure the distribution characteristics.

In order to understand the seasonal distribution characteristics of eddies in the Bay of Bengal more intuitively, we used monthly averaged SLA fields to identify eddies that occur frequently in certain regions (here we call them "the monthly eddy"). For such monthly eddies, the tracking processing is not performed, and these monthly results identified from monthly SLA fields are processed as individual eddies. Each individual monthly eddy is counted as one eddy. A monthly eddy may not be the same eddy in the continuous daily SLA fields or may be an average of many eddies passing on that region in one month. But it doesn't matter, we're trying to determine areas where eddies often exist, and to study the spatial distribution characteristics of eddies. It is worth noting that we only use "the monthly eddy" result in Table 1 and Figure 3 in this section to study the seasonal spatial distribution of eddies. In the remainder, e.g., the study of eddy vertical structure (Section 3.2) and the heat-salt transport (Section 4), the eddy trajectory result from daily SLA fields are used.



As a result, a total of 1351 CEs and 1190 AEs (individual monthly eddies) were identified from the monthly SLA fields (Table 1). The statistical analysis of these monthly eddies shows that the mean amplitudes of CEs and AEs are both about 8.3 cm, and the mean radii are 125 km and 136 km, respectively. Moreover, the eddy amplitude and radius vary significantly in different seasons. Eddy amplitudes are higher in spring and summer than in autumn and winter, especially for AEs in spring and CEs in summer which mean amplitudes are close to 10 cm. The number of eddies with amplitudes ≥ 10 cm also shows similar seasonal changes, indicating that eddy intensities in spring and summer is relatively strong, and that in autumn and winter is relatively weak. In terms of eddy radius, mean radii of CEs in summer and autumn are larger than in winter and spring; mean radii of AEs in spring and summer are larger than in autumn and winter. Based on these monthly eddy results, eddies are classified according to different seasons, and the seasonal spatial distribution of CEs and AEs are given in Figure 3. In order to understand the influence of circulations in the Bay of Bengal on mesoscale eddies, Figure 4 shows the 26-year monthly averaged SLA and geostrophic current anomaly fields in the Bay of Bengal.

Table 1. The monlthly eddy number and mean properties in different seasons.

|  |  | Winter | Spring | Summer | Autumn | Total |
|---|---|---|---|---|---|---|
| **Number** | CEs | 319 | 366 | 439 | 227 | 1351 |
|  | AEs | 329 | 298 | 392 | 171 | 1190 |
| **Number with Amplitude ≥ 10 cm** | CEs | 43 | 106 | 159 | 59 | 367 |
|  | AEs | 71 | 107 | 116 | 30 | 324 |
| **Amplitude/cm** | CEs | 6.5 | 8.6 | 9.5 | 7.9 | 8.3 |
|  | AEs | 7.8 | 9.5 | 8.6 | 6.9 | 8.3 |
| **Radius/km** | CEs | 122 | 117 | 130 | 135 | 125 |
|  | AEs | 124 | 137 | 148 | 128 | 136 |

It can be seen from Figure 3 that CEs and AEs have obvious seasonal variation in their local distribution characteristics. In the Winter monsoon season (Figure 3 a and e), although eddies are distributed throughout the Bay of Bengal, many CEs with strong amplitude and large radius are clustered in western parts of the bay, while many high-intensity AEs (amplitudes >15 cm) are concentrated in northern parts. The monthly averaged SLA fields in winter (Figure 4) indicate that from October to January of the following year, the bay is dominated by a clear cyclonic gyre, accompanied by abundant CEs in the western bay. During this period, the EICC flows southward along the western coast of the bay, which also contributes to stable development of these CEs in the western bay. Meanwhile, there are areas of high SLA in the northern portion of the bay (about 87°E, 18°N), and a persistent AE forms there in January. Subsequently, the AE continues to develop and grow, and deforms the cyclonic gyre, eventually causing reversal of the northern EICC from its original southward flow to a northward flow in February (Supplementary Material Figure S1). The northern EICC reversal occurs prior to the monsoon transition, and it can reach a northeastward speed of > 0.5 m·s⁻¹ (Potemra et al., 1991; Yu et al., 1991). This early reversal of the EICC in winter is related to the strong AE in the northern bay. It shoud be noted that low-amplitude large-scale CEs often occur in the low-latitude equatorial regions in winter. These CEs, which generally appear in the western waters of Sumatra, are largely manifestations of Rossby waves and move gradually westward or northwestward with the westward drift of the monsoon (see eddy trajectories in Supplementary Material Figure S3).



In the Spring premonsoon season, many small but high-strength CEs are concentrated in northernmost and western parts of the bay, while AEs are concentrated in western and northwestern parts (Figure 3 b and f). The monthly averaged SLA fields in spring (Figure 4) indicate that from March to May, a basin-scale anticyclonic gyre appears and dominates the

270 bay. Within the anticyclonic gyre, many strong AEs apprear in the western and northwestern parts of the bay. A cyclonic structure is prone to appear in the northernmost part of the bay, owing to river runoff and coastal current baroclinic instability (Patnaik et al., 2014; Babu et al., 2003; Kumar and Chakraborty, 2011). As river runoff increases, the CE continues to grow and strengthen (amplitudes >15 cm), causing the flow of the northern EICC to become unstable and to separate from the coast at around 18°N (Supplementary Material Figure S1). Concurrently, owing to the seasonal variation of the EICC, CEs

and AEs often appear alternately in western parts of the bay.

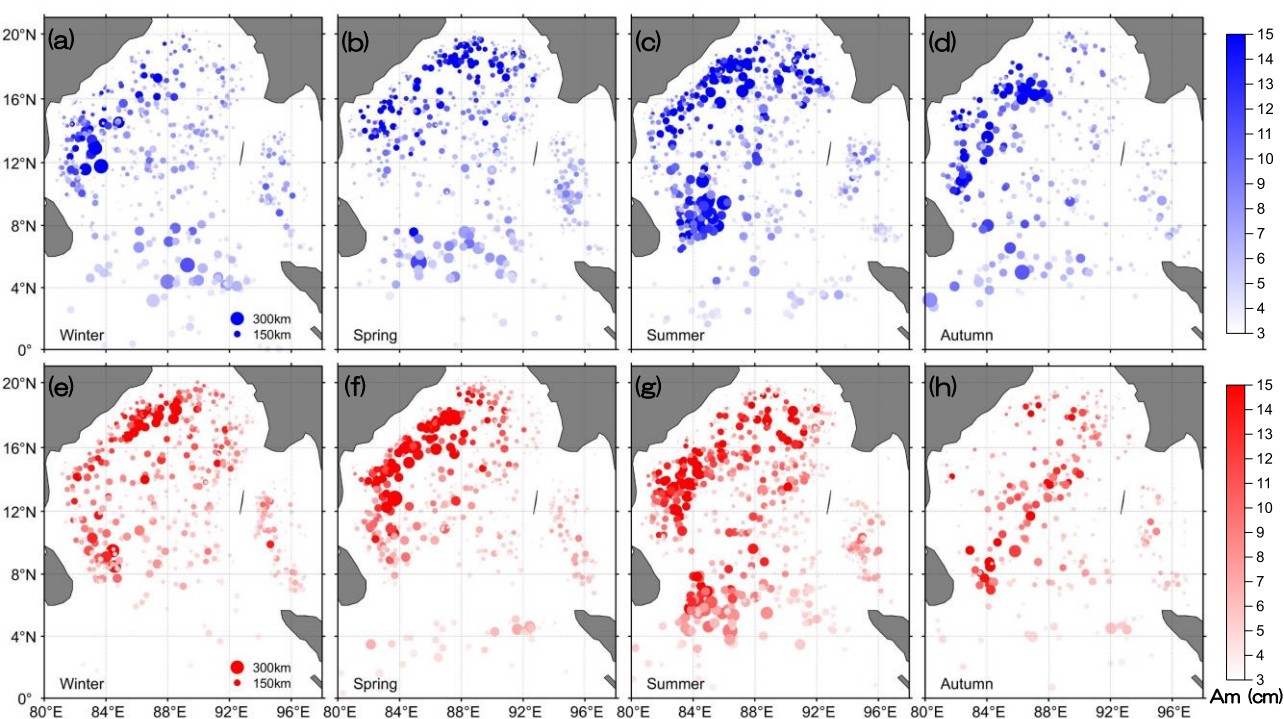

Figure 3: Seasonal spatial distribution of monthly cyclonic eddies (upper) and anticyclonic eddies (lower) based on monthly-averaged sea level anomaly fields from January 1993 to February 2019 in the Bay of Bengal. Blue and red points represent cyclonic eddies and
280 anticyclonic eddies, respectively, where the color intensity represents the eddy amplitude (Am, unit: cm), and the size of the point represents eddy scale (radius).



Figure 4: The 26-year monthly averaged sea level anomaly (SLA, color shading) and geostrophic current anomaly (arrows) fields in the Bay of Bengal.

In the Summer monsoon season, CEs are distributed mainly in northwestern and northeastern parts of the bay, and to the east of Sri Lanka, while AEs are distributed primarily in the eastern and western parts, and to the southeast of Sri Lanka (Figure 3 c and g). Throughout the Summer monsoon, the upper-level circulation structure is rather chaotic, and the EICC also becomes changeable. Some persistent CEs often appear on the northern side of the EICC, while AEs are often shed on its southern side. Many CEs and AEs are generated near the eastern boundary, propagate southwestward, and are clustered in the eastern and northeastern parts of the bay (Supplementary Material Figure S3). These eddies are mainly driven by equatorial zonal winds, with both nonlinearity and coastline geometry essential for eddy generation (Cheng et al., 2018). It is noteworthy that a large number of high-intensity CEs are concentrated in the eastern seas of Sri Lanka in the Summer monsoon season, while corresponding AEs often appear in the south. Since the Southwest Monsoon is obstructed by the land




area of Sri Lanka, there is a strong positive wind stress curl to the east of Sri Lanka, which is conducive to the formation of the cyclonic structure (Murty et al., 1992; McCreary et al., 1996; Kurien et al., 2010). Moreover, the Southwest Monsoon Current outside of the bay flows southeastward and invades the bay through the east of Sri Lanka, which increases the strength of the cyclonic structure further. Therefore, a presistant and strong CE with very fast rotation and high eddy kinetic energy often appears to the east of Sri Lanka (refers to the Sri Lanka cold eddy). Similarly, to the south of the Sri Lanka cold eddy, a compensatory anticyclonic structure is produced owing to the variation of intrusive currents and driven by the monsoon.

In the Autumn postmonsoon season, the strength of the Southwest Monsoon decreases rapidly in October to be replaced by the Northeast Monsoon in November, and a basin-scale cyclonic gyre is formed throughout the entire bay. Together with the southwestward EICC, many CEs appear in northwestern and western parts of the bay, whereas there are few AEs (Figure 3 d and h). The Sri Lanka cold eddy generated in summer moves northward and interacts with the reversed EICC before finally disappearing in the western parts of the bay (Supplementary Material Figure S3). Consequently, the southward velocity of the EICC increases markedly. The AE to the southeast of Sri Lanka also moves northward, forming a relatively concentrated distribution in the water to the east of Sri Lanka. In addition, in the central bay, some AEs often appear accompanied by local current variations.

Analysis on the seasonal spatial distribution of individual monthly eddies from monthly SLA fields revealed that mesoscale eddy activity in the Bay of Bengal has evident seasonal variation, which has notable impact on the circulation system within the entire bay. Generally, there are three main areas of distribution of mesoscale eddies in the Bay of Bengal. One is the EICC region in the west and northwest of the bay, indicating that variation or reversal of the western boundary current EICC will often shed rich eddy structures, especially in spring and summer. Another region is the northeastern part and the eastern boundary, where eddies generated in spring and summer move southwestward into the central bay in autumn (some even reach the western bay). The third region is seas to the east of Sri Lanka, where there are strong CEs in summer and AEs in autumn.

The heat and salt transport efficiencies of mesoscale eddies are related closely to eddy propagation speed (Dong et al., 2014; Gonaduwage et al., 2019; Stammer, 1998). In order to study the propagation direction and speed of mesoscale eddies in the Bay of Bengal, all eddy trajectories with lifetime ≥30 days from daily SLA fields are analyzed here. The average speed of propagation of eddies in the Bay of Bengal is shown in Figure 5. In general, eddies move slowly in the western and northern parts of the bay and move faster in central and southern parts. The averaged zonal component $u$ of eddy propagation speed (Figure 5b) shows that the westward speed of eddies gradually increases from <5 cm·s$^{-1}$ in the north to up to 20 cm·s$^{-1}$ in low-latitude equatorial regions. Since eddy generation is often associated with nonlinear processes, there are apparent discrepancy between eddy propagation and the phase speed of the first baroclinic Rossby waves (Chen et al., 2012; Chelton et al., 2011a). In addtion, eddy propagation speeds clearly bounded by the 12°N line of latitude; eddies to the north move westward and slightly southward, while eddies to the south move westward and slightly northward. In terms of the averaged meridional component $v$ of eddy propagation speed, the value to the north of 12°N is generally negative (southward), while $v$ to the south of 12°N is largely positive (northward). The eddy propagation speed in different seasons also shows some differences. The westward speed of eddies is fastest in winter, followed by spring and autumn, and slowest in summer (Figure 5b). The meridional speed of eddies is faster in winter and spring than in summer and autumn, and eddies move





southward outside the bay in summer and autumn (Figure 5c). The seasonal propagation speed of eddies may be related to the seasonal variation of the overall circulation, especially in the southern part of the bay to the south of 12°N. In winter, the

background current is basically westward/southwestward, which is more conducive to the westward propagation of eddies; while in summer, the drifting intrusion of the Southwest Monsoon Current blocks the westward movement of eddies. For the southern bay, eddies in winter and spring mainly appear outside the bay and move northwestward (see Supplementary Material Figure S3), therefore the northward speed is larger. While some eddies in summer and autumn appear in southeast bay and move southwestward, which counteracts the northward speed of the Sri Lanka cold eddy. The different directions

and speeds of propagation of eddies in different seasons are crucial to estimation of the magnitude of the seasonal transport of eddies in the Bay of Bengal.

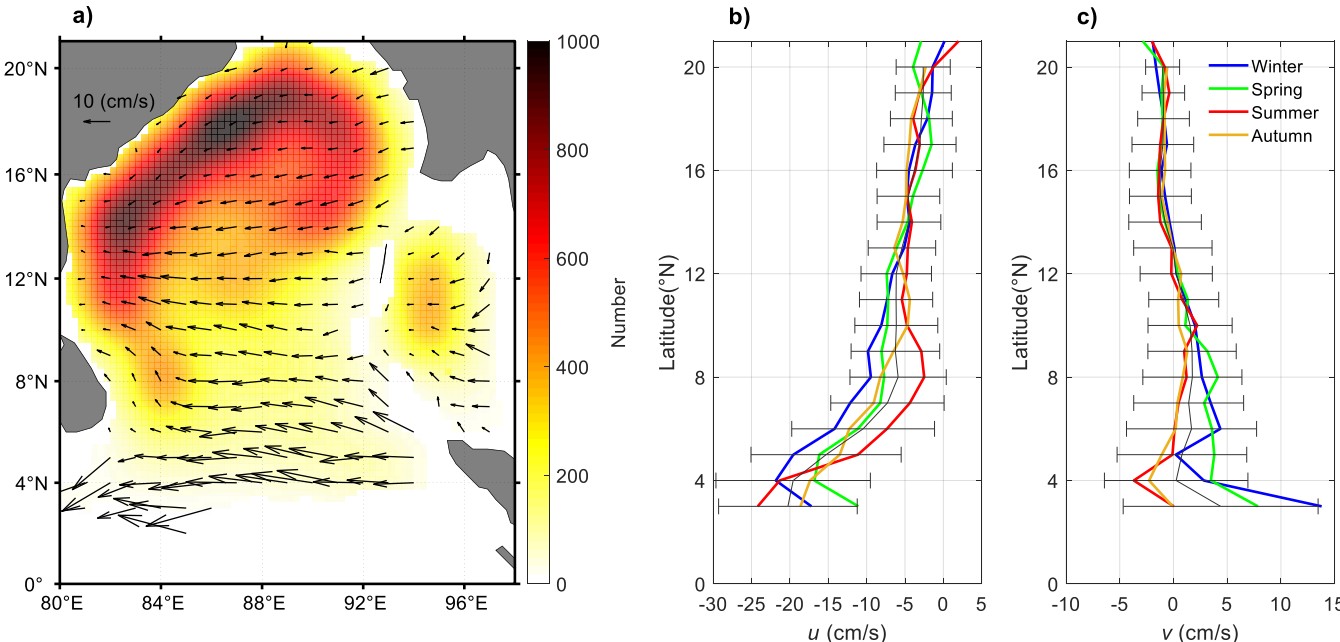

Figure 5: (a) Statistical eddy propagation speed (color indicates the numbers of eddy interiors for eddy trajectories with lifetimes ≥ 30 days that passed through each 1°×1° region), and (b) the zonal component *u* and (c) the meridional component *v* (thin solid line represents the mean value and its standard deviation, and the color represents the season) in the Bay of Bengal based on daily SLA fields spanned a 26-year period from January 1993 to February 2019.

## 3.2 Seasonal variation of vertical thermohaline structure of eddies

In the previous section, the statistical analysis of the mesoscale eddy activity in the Bay of Bengal are preformed based

on 26-year period SLA fields, and the result shows that the eddy activity has obvious seasonal variation due to complex exogenous effects. To reveal the seasonal variation of the vertical thermohaline structure of eddies in the Bay of Bengal, the 3D thermohaline structures of the eddies were constructed by surfacing the Argo float profiles into SLA-based eddy areas. In this study, all eddy trajectories with lifetime ≥30 days from daily SLA fields and Argo profiles choosed following Chaigneau et al. (2011) were used for eddy composition. We considered the detection results (from daily SLA fields) of the long-lived

eddy trajectory to match the Argo profiles on the same day, and selected Argo profiles with a distance of <1.5 radii from the





eddy core for vertical eddy structure analysis. The Argo profiles acquired within eddies were classified according to season. In addition, to verify the vertical thermohaline structure obtained from Argo profiles, weekly 3D temperature and salinity field data from the ARMOR3D reprocessed dataset were used to match surface eddies identified from daily SLA fields at the closest time. Such that we could also obtain the 3D temperature and salt structure of each eddy from weekly 3D temperature and salinity field data.


The regional water masses play an important role on temperature and salinity structures of eddies. Based on all Argo profile data in the Bay of Bengal, we calculated the seasonally averaged temperature and salinity profiles (Figure 6). At the surface and thermocline of the ocean (about upper 300 dbar), seasonal changes of the water masses are evident. In the low-latitude equatorial area, the surface Arabian Sea High-Salinity Water (ASHSW) characterized by a temperature of 24−30°C and salinity of 34.5−36 psu, flows from the Arabian Sea to the Bay of Bengal (Emery and Meincke, 1986). The


Bay of Bengal is characterized by surface low-salt Bay of Bengal Water (BBW) with a salinity of <34 psu (even <30 psu in the far north bay). The BBW is the low temperature and less salinity in winter, and the high temperature and less salinity in autumn. In the deep sea, water masses are mainly dominated by the Indian Equatorial Water (IEW), characterized by a temperature of 10−23°C and salinity of 34.5−35 psu (Stramma et al., 1996), with less seasonal variation. Considering the


hydrological differences from north of the bay to the south, here the Bay of Bengal is divided latitudinally into north and south subregions with 12°N as the boundary to study the eddy 3D structure of each subregion.

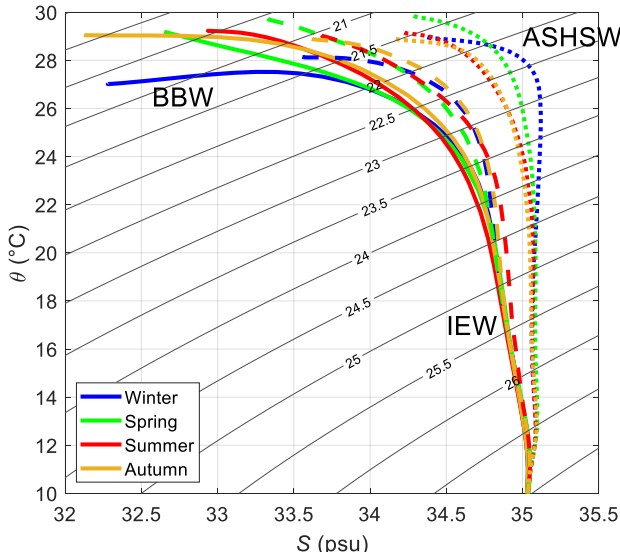

Figure 6: Mean potential temperature and salinity ($\theta$-$S$) diagram calculated for different seasons (colored lines) based on all Argo profiles obtained in the Bay of Bengal. The solid, dashed and dotted lines represent the mean $\theta$-$S$ profiles in the region north of 12°N, 6°−12°N and

0°−6°N, respectively, and the gray contours denote potential density $\sigma$ (kg·m$^{-3}$). BBW, the surface Bay of Bengal Water; ASHSW, the Arabian Sea High-Salinity Water; IEW, the Indian Equatorial Water.

Based on Argo profiles matched with eddies in different seasons, the mean vertical profiles of the potential temperature anomaly $\theta'$ and salt anomaly $S'$ of eddies in the Bay of Bengal were shown in Figures 7 and 8. It can be seen that there are

obvious seasonal variations in the temperature and salinity anomalies of the eddies, as well as some differences for the northern and southern bay. Specifically, for $\theta'$ caused by eddies in the northern bay (upper panels in Figure 7), the negative




(positive) extrema of CEs (AEs) are located at approximately 100 dbar (120 dbar) due to the water body within eddies uplifts (sinks) the thermocline. The $\theta'$ of CEs and AEs are both maximum in spring, up to ±2.5°C, and minimum in winter, about ±1.2°C, and about ±2°C in summer and autumn, respectively. The minimum $\theta'$ in winter is not only related to the weaker

eddy intensity (Figure 3a and e), but also to the thicker barrier layer in this season (Thadathil et al., 2007), which attenuates temperature changes through vertical movement of the water body. In spring, the barrier layer becomes thinner, and the vertical movement of the water body becomes easier, so the temperature anomaly is largest during the season. For $\theta'$ caused by eddies in the southern bay (lower panels in Figure 7), the negative (positive) extrema of CEs (AEs) are located at approximately 80 dbar (100 dbar), which is shallower than that in the northern bay due to the shallower thermocline in the

southern bay (Figure 6). The $\theta'$ of CEs is the largest in summer, reaching −3°C, the smallest in winter, less than −2°C; the $\theta'$ of AEs is larger in summer and autumn, around +2°C, and smaller in winter and spring, around +1.5°C. The maximum $\theta'$ of CEs in the southern bay in summer is associated with the presistant and strong Sri Lanka cold eddy which often appears in May and disappears in August (Figure 3c). In general, comparison with some western boundary current regions (Chaigneau et al., 2011; Yang et al., 2013), the depth of influence of eddies in the Bay of Bengal is relatively shallow and generally

limited to the upper 300 dbar (the deepest effect does not exceed 500 dbar) due to fewer high-intensity eddies and the stratification of the surface waters (Cui et al., 2021).

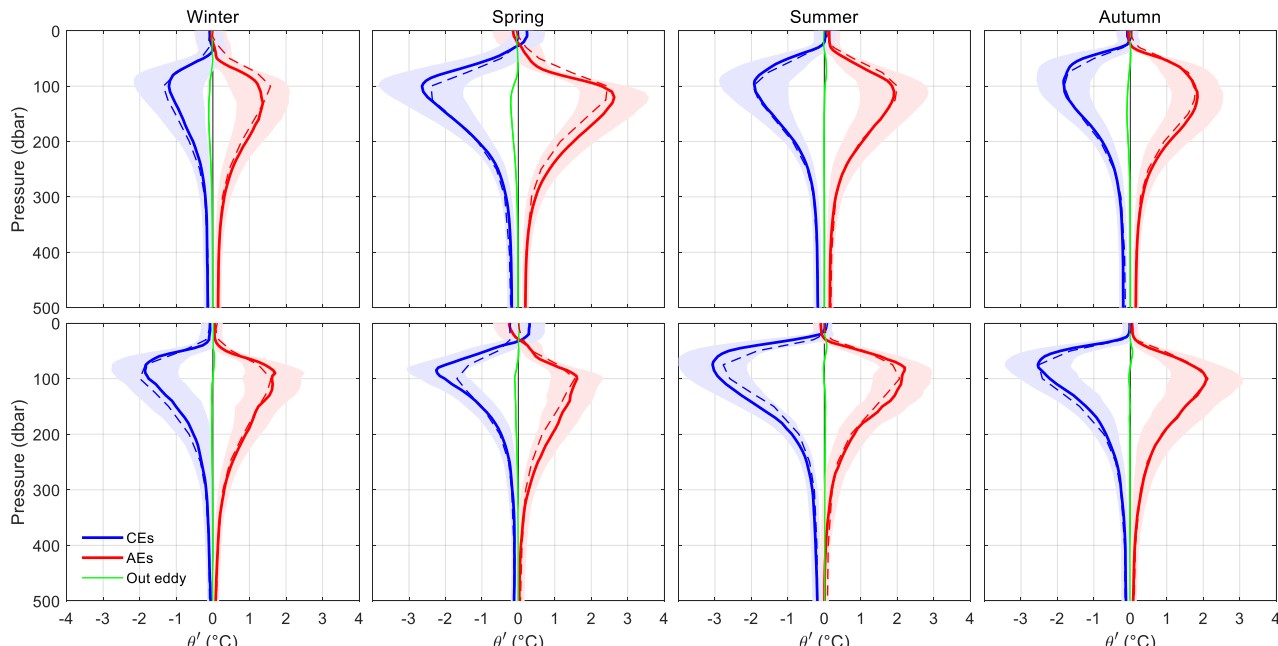

Figure 7: Mean vertical profiles of the potential temperature anomaly $\theta'$ of composite cyclonic eddies (CEs, blue lines) and anticyclonic eddies (AEs, red lines) in different seasons for the northern (upper) and southern (lower) bay. The green lines indicate the mean anomalies

that were computed from Argo profiles outside eddies relative to the Argo seasonal-mean climatologic profiles. The solid lines indicate the mean anomalies from Argo profiles, the shading indicates the range of one standard deviation, and the dashed lines indicate the mean anomalies from the weekly ARMOR3D temperature and salinity field data.



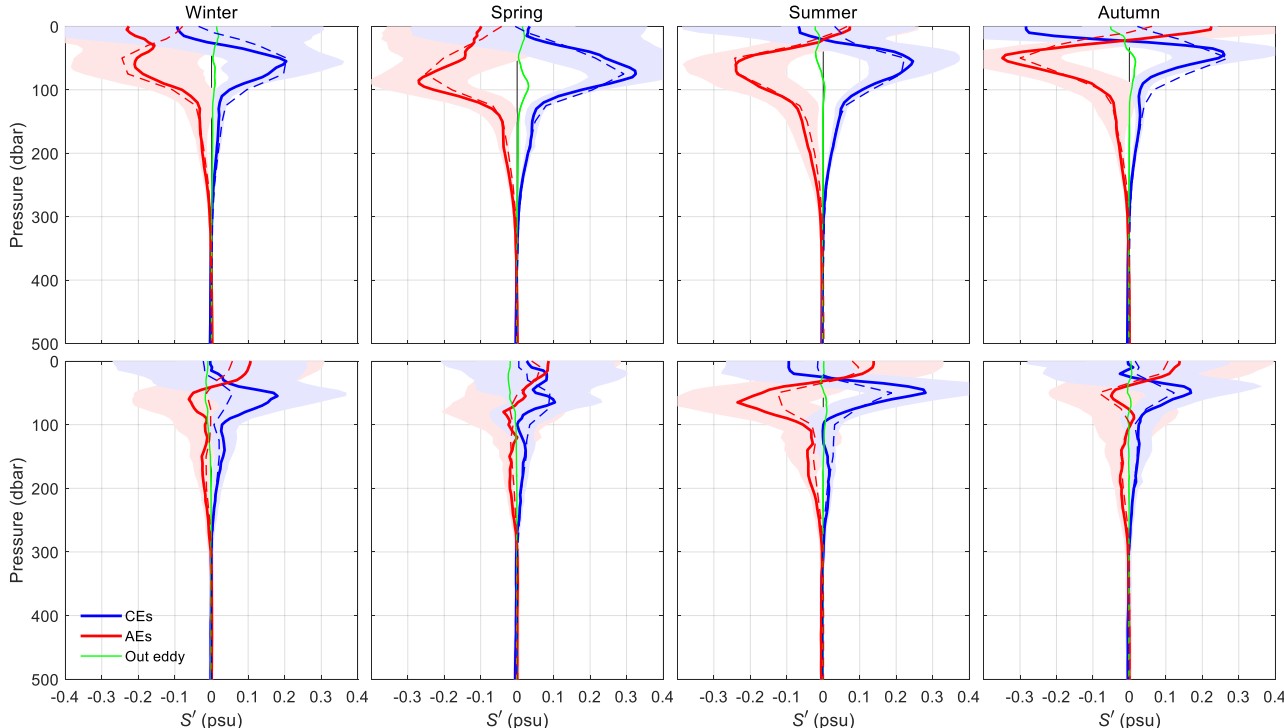

Figure 8: Same as Figure 7 but for salinity anomaly $S'$.


The vertical salinity structures inside the eddies in different seasons for the northern and southern bay are illustrated in Figure 8. Salinity anomalies caused by eddies in the Bay of Bengal are limited to the upper 300 dbar and exhibit obvious seasonal variation in the upper 100 dbar surface water. Compared with the vertical temperature anomalies $\theta'$, the salinity anomalies $S'$ of eddies in the north and south bay present larger differences. For $S'$ in the northern bay (upper panels in Figure 8), CEs (AEs) produce notable positive (negative) signals at the subsurface. The maximum $S'$ of CEs in spring can exceed +0.3 psu, whereas it is around +0.25 psu in autumn, and weakest in winter is only +0.2 psu; the corresponding depth is generally in the range of 50−80 dbar (spring with the maximum depth of about 80 dbar, autumn with the shallower depth of about 50dbar). The extremum of negative $S'$ in AEs in autumn can reach −0.35 psu, values are around −0.25 psu in spring and summer, and −0.2 psu in winter. The corresponding depth with extremum of negative $S'$ in AEs is nearly 100 dbar in spring, 50−60 dbar in summer and autumn, and at the surface in winter. The $S'$ of CEs and AEs in the 30 dbar shallow surface water in summer and autumn exhibit some perturbation (positive/negative signals in CEs/AEs). For $S'$ in the southern bay (lower panels in Figure 8), the magnitude of $S'$ signals is significantly small. In summer, the $S'$ of CEs and AEs are both maximum, exceeding ±0.2 psu. In other seasons, the $S'$ of CEs and AEs are small, with the magnitude of less than 0.1 psu. The weaker salinity anomaly in eddies in the south bay is partly related to the fewer and weaker eddies there. There are few concentrated and strong eddies in the south bay in winter, spring and autumn, while the strong Sri Lanka cold eddy and a corresponding warm eddy apprear in summer (Figure 3), so the salinity anomaly in eddies is remarkable in this season. On the other hand, due to the small vertical gradient of salinity in the southern bay (Figure 6), the vertical movement of the water body caused by eddies is difficult to cause a large salinity change.





In comparison with the seasonal variation of $\theta'$ caused by eddies, the salinity structure exhibits some complex disturbance signals. However, below the surface, salinity variation caused by eddies in different seasons (limited to the upper 300 dbar) is evident and related primarily to the properties of eddies in each season. On the whole, under the control of the low-salinity BBW at the surface and the IEW in the deep ocean, the Bay of Bengal presents the salinity structures of positive $S'$ inside CEs and negative $S'$ inside AEs in the thermocline.

The weekly ARMOR3D temperature and salinity field data were also used to analyze the seasonal variation of the vertical thermohaline structure of the eddies, and the corresponding results are drawn by dashed lines in Figures 7 and 8. The seasonal variations of temperature and sanility signals are largely consistent with the result for the composite eddies based on the Argo profiles. This shows that the weekly ARMOR3D thermohaline field data could also be used to study the thermohaline structure of eddies in the Bay of Bengal. We matched the eddy results identified from daily SLA fields with the weekly ARMOR3D temperature and salinity field data at the closest time such that we could obtain the 3D temperature and salt structure of each eddy. Considering the full spatial coverage of the data, the spatial characteristics of the vertical thermohaline structure of the eddies in the Bay of Bengal was analyzed (Figure 9). To ensure the accuracy of the data, we only calculated the average temperature and salt anomalies of subsurface water within the eddies (i.e., 50−150dbar, which is the depth layer where eddies cause the greatest variations in temperature and salinity).

The spatial distribution of eddy-induced temperature anomalies is largely the same as the seasonal spatial distribution of eddies shown in Figures 3 and 4, i.e., in areas where there are more strong eddies, the temperature anomaly is generally larger. In winter, the CEs concentrated in the western bay correspond to a variation of $\theta'$ of approximately −2°C, while the AEs concentrated in the northwestern bay correspond to an obviously positive $\theta'$. The corresponding characteristic in spring is more obvious, i.e., there are alternating CEs and AEs in the northwestern bay that correspond to obvious positive and negative variation of $\theta'$, respectively. In summer and autumn, the concentration of CEs in the northwestern bay and to the east of Sri Lanka cause significant negative $\theta'$, especially in the area to the east of Sri Lanka where the value of $\theta'$ can exceed −3°C. Similarly, AEs cause high positive values of $\theta'$ in the western bay in summer and to the east of Sri Lanka in autumn. This shows that the vertical temperature structure of eddies in different regions of the Bay of Bengal is notably different owing to the difference in eddy distribution. If the average temperature structure of the entire region were used to estimate the eddy-induced heat transport, the marked regional characteristics would be smoothed. Therefore, accurate estimation of eddy-induced heat transport should consider the temperature anomaly of each individual eddy, together with its direction and speed of propagation, such that the regional characteristics of heat transport might be reflected.

The spatial distribution of eddy-induced salinity anomalies is more complicated than that of temperature. In the region of the Bay of Bengal to the north of 12°N, the basic characteristics of CEs correspond to positive salinity anomalies, while those of AEs correspond to negative salinity anomalies. In the southern part to the south of 12°N, the salinity signal becomes turbulent owing to the invasion of the low-latitude equatorial circulation. For example, AEs present confused positive salinity anomalies in the southern bay. Owing to differences in the salinity anomaly signal between the northern and southern parts of the bay, confused anomaly signals will appear in the surface during analysis of the 3D structure of one eddy in the entire Bay of Bengal (Figure 8; Lin et al., 2019; Gulakaram et al., 2020). Some studies suggested that this reflects a salinity dipole structure in the near surface layer due to the horizontal advection, eddy rotation and background temperature/salinity meridional gradient (Melnichenko et al., 2017; Amores et al., 2017).





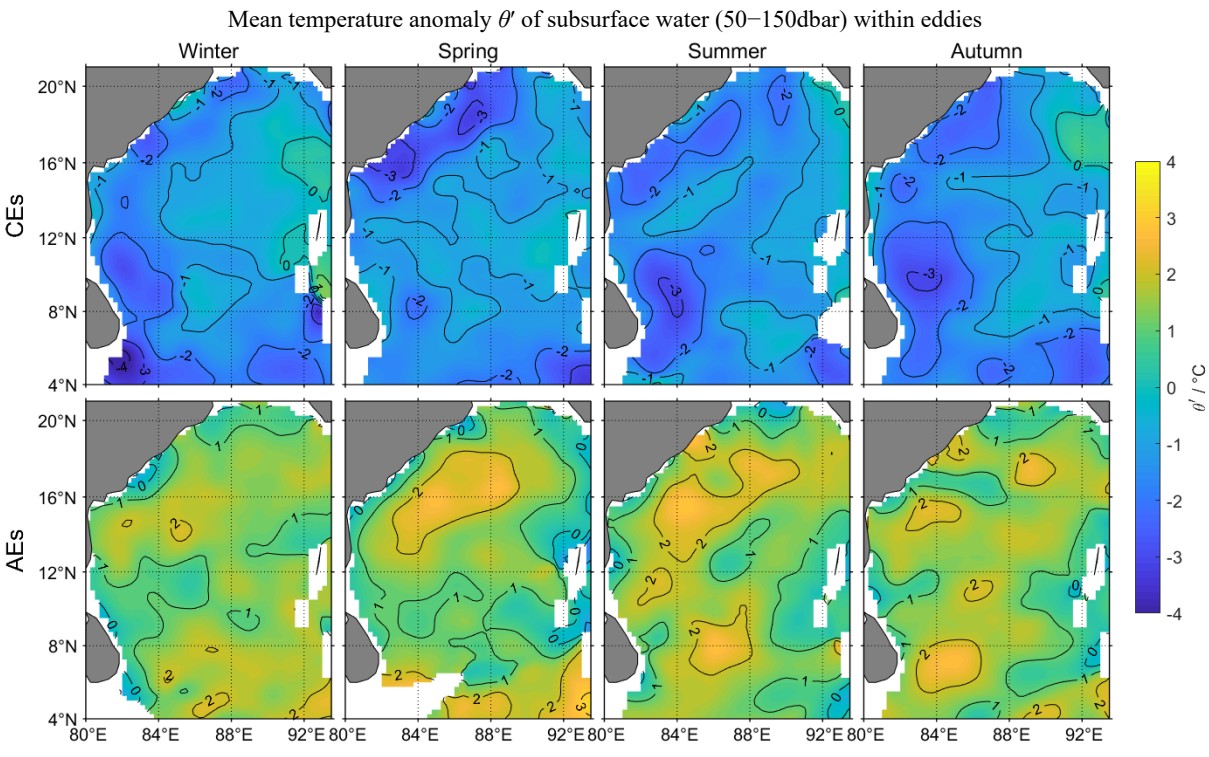

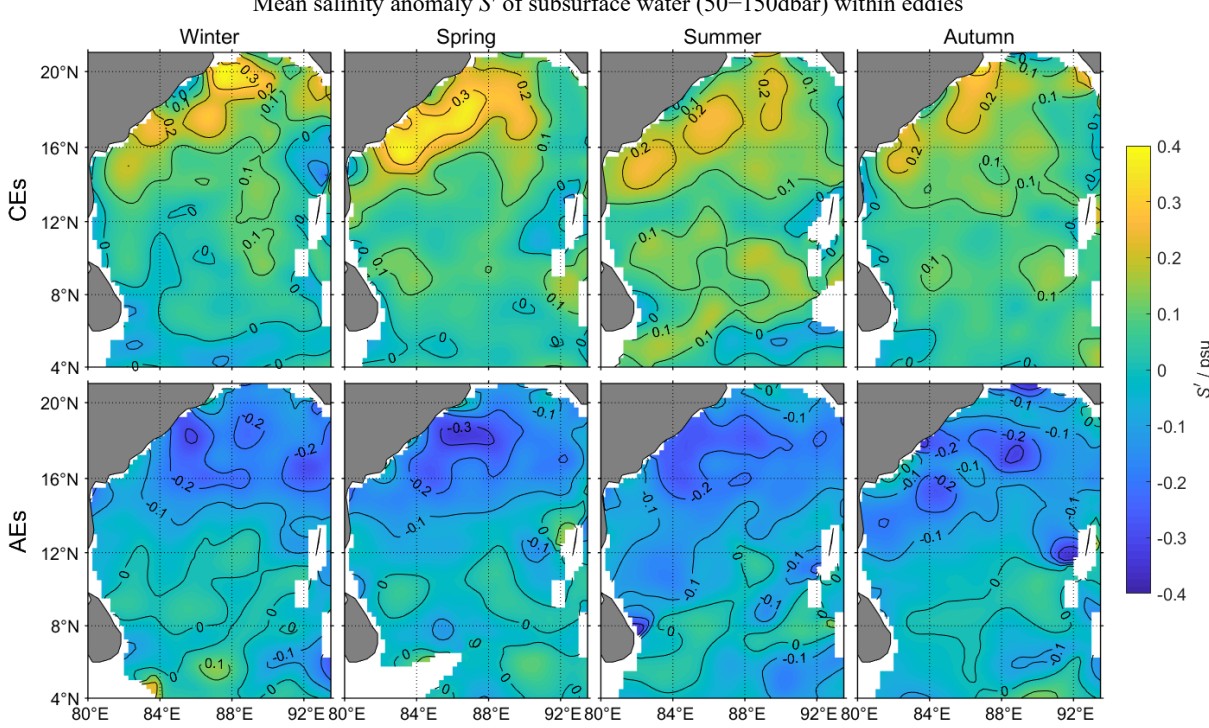

Figure 9: Spatial characteristics of the vertical temperature and salinity anomalies of cyclonic eddies (CEs) and anticyclonic eddies (AEs) in the Bay of Bengal in different seasons based on the weekly ARMOR3D temperature and salinity field data.



## 4 Seasonal eddy-induced heat and salt transports in the Bay of Bengal

Eddy heat transport is traditionally estimated within a Eulerian framework (Qiu and Chen, 2005; Roemmich and Gilson, 2001; Stammer, 1998), which does not explicitly identify eddy movements. Although Dong et al. (2014) presented the global eddy heat and salt transport in large ocean basins combining eddy data with Argo profiles, the detailed spatial distribution maps of transports are obscure. Many studies (Castelao, 2014; Chaigneau et al., 2011; Schütte et al., 2016) focused on regions where eddies dissipate to calculate the eddy-induced ocean heat gain or loss, based on the assumption that eddies release the heat and salt anomalies trapped in their cores when they dissipate. However, eddies will exchange heat and salt with the background current during their movements. In this study, similar to Dong et al. (2017), we considered changes in eddy structure along the paths of eddy propagation to estimate the eddy-induced heat and salt transport in the Bay of Bengal. Therefore, by combining the temperature and salinity anomalies of each eddy along the eddy path, provided by the weekly ARMOR3D temperature and salinity field data, with the details of eddy movement (propagation trajectory), provided by daily SLA fileds, we estimated the eddy-induced heat and salt transport in different areas of the Bay of Bengal. The detailed method, as described in Section 2.2, considers not only the direction and speed of eddy propagation, but also the variation of the properties of the intrinsic heat and salt during eddy movement.

### 4.1 Eddy-induced heat transport and its seasonal variation

The seasonal heat transport attributable to mesoscale eddies in the Bay of Bengal is illustrated in Figure 10. It can be seen that CEs/AEs almost present eastward/westward heat transport due to CEs/AEs generally carry negative/positive heat anomalies westward across the bay (upper and middle panels). The heat transport associated with CEs and AEs jointly determines the spatial distribution of heat transport of all the eddies in the entire Bay of Bengal (lower panels). Owing to differences in the spatial distribution of mesoscale eddies in the Bay of Bengal (Figure 3), the eddy-induced heat transport is generally higher in regions where eddies are concentrated. In the area to the south of 8°N, the southeast outside of the bay, the high eastward heat transport in winter and spring is related to the large-scale CEs that often appear there and move westward at a high speed (e.g., >10 cm·s⁻¹, Figure 5). The seas to the east of Sri Lanka are dominated alternately by CEs and AEs in different seasons. Thus, in this region, the directions of heat transport are different in different seasons (e.g., in autumn and winter, westward-moving AEs lead to westward heat transport; in summer, northward-moving CEs lead to southward heat transport), and the magnitude of this transport is generally >15×10⁶ W·m⁻¹. The western bay is dominated by CEs and presents eastward heat transport in autumn and winter; conversely, it is dominated by AEs in spring and summer and presents westward heat transport. In the area of the EICC area (i.e., the northwest of the bay), owing to the combined action of CEs and AEs, there is notable northeastward heat transport only in summer and autumn. In the eastern bay, the prevalence of AEs moving westward in autumn and winter (Cheng et al., 2018; also see Figure 3), it generally corresponds to westward heat transport.





Figure 10: Seasonal eddy-induced heat transport $Q_h$ in the Bay of Bengal: (upper) results for cyclonic eddies, (middle) results for anticyclonic eddies, and (lower) results for all eddies. Here, $Q_h = (Q_{hm}, Q_{hz})$ is a vector whose components are the meridional and zonal heat transports, the arrows indicate the transport direction, and the color indicates the transport magnitude. The four subregions (NB, the northern bay; WB, the western bay; CB, the central bay; SB, the southern bay) and 5 sections (S1 and S2, 16°N; S3 and S4, 10°N; S5, 84°E) of the Bay of Bengal are given in the lower right panel.

In order to study the eddy-induced heat transport in different sub-regions of the Bay of Bengal, the bay is divided into 4 subregions by 5 cross-sections (the lower right panel in Figure 10). We integrated the zonal heat transport $Q_{hz}$ by mesoscale eddies at each 0.25° grid from north to south, and obtained the integrated zonal heat transport $ZHT = \int Q_{hz} dy$ in the entire meridional direction, where $dy$ is the meridional unit distance (unit: m) such that the unit of $ZHT$ is Watts (abbr. W). Similarly, the zonally integrated meridional heat transport $MHT$ can be expressed as $MHT = \int Q_{hm} dx$, where $Q_{hm}$ is the meridional heat transport and $dx$ is the zonal unit distance. The seasonal $ZHT$ and $MHT$ caused by eddies within the 4 subregions and the whole bay, as well as the heat transport across the 5 sections are presented in Figure 11.

The magnitude of the seasonal $ZHT$ of CEs and AEs in the whole bay is in the order of $10 \times 10^{12}$ W, with higher values in autumn and winter and smaller values in spring and summer. Because the $ZHT$ directions of CEs and AEs are opposite, the



*ZHT* of all eddies in the whole bay is small, and the value in each season is below 2×10$^{12}$ W. In terms of sub-regions, the seasonal *ZHT* of all eddies in NB and SB are small, and they both show weak eastward heat transport. The eastward *ZHT* in

WB is high in autumn and winter, exceeding 5×10$^{12}$ W, and the westward *ZHT* in spring and summer is about 2×10$^{12}$ W. The *ZHT* in CB is westward in all seasons, and the values are high in autumn and winter, about 3−4×10$^{12}$ W. The magnitude of the seasonal *MHT* of all eddies in the whole bay is small, mostly below 1×10$^{12}$ W. The *MHT* in the subregions differs significantly. The NB presents high northward *MHT* in spring, summer and autumn, reaching a maximum of 4×10$^{12}$ W in summer. The *MHT* in WB and CB is small with northward heat transport in WB and southward heat transport in CB. The

seasonal variation of *MHT* in SB is the most obvious, with high southward *MHT* in spring and summer (approximately 5×10$^{12}$ W in spring) and northward *MHT* in autumn (exceeding 4×10$^{12}$ W).

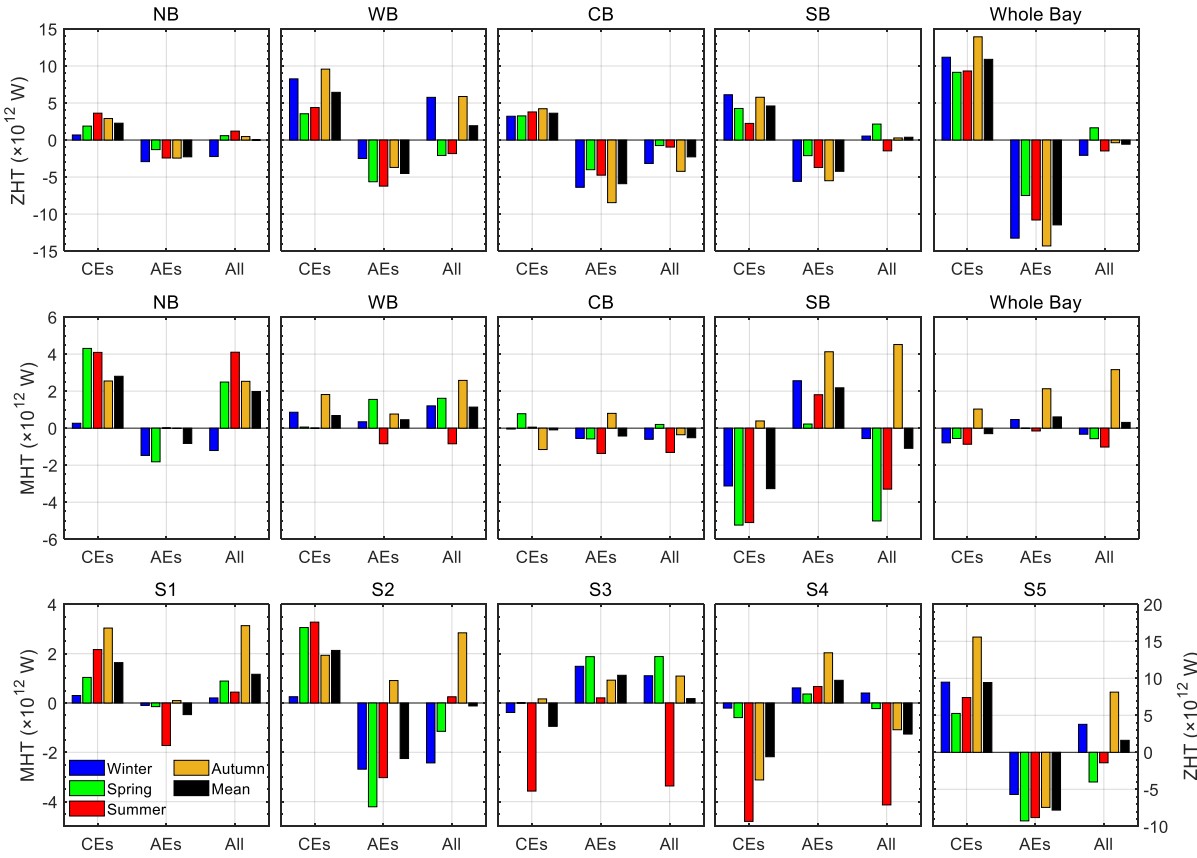

Figure 11: The meridionally integrated zonal heat transport (*ZHT*, upper) and zonally integrated meridional heat transport (*MHT*, middle)
by cyclonic eddies (CEs), anticyclonic eddies (AEs), and all eddies (All) in different seasons for 4 subregions and the whole Bay of Bengal. (lower) The seasonal heat transport by eddies across 5 sections. The four subregions and 5 sections are shown in the lower right panel in Figure 10.

The heat transport across 5 cross-sections represents the eddy-induced heat exchange between the different subregions
(lower panels in Figure 11). The positive *MHT* across S1 section indicates that the eddy carries heat from the WB region northward into the NB region, the value is maximum about 3×10$^{12}$ W in autumn. The S2 section is the exchange channel between NB and CB, across which the *MHT* is southward (from NB to CB) in winter and spring and northward (from CB to



NB) in autumn. The *MHT* between WB and SB (across S3 section) is southward (from WB to SB) with a value of $>3\times10^{12}$ W in summer and northward (from SB to WB) in other seasons. The S4 section exhibits nearly $4\times10^{12}$ W southward *MHT* (from CB to WB) in summer, the *MHT* in other seasons is relatively small. The *ZHT* between WB and CB (across S5 section) is eastward (from WB to CB) in autumn and winter with a maximum of approximately $8\times10^{12}$ W in autumn, and westward (from CB to WB) in spring and summer.

The seasonal eddy-induced *ZHT* at different longitudes and *MHT* at different latitutdes in the whole Bay of Bengal are shown in Figure 12. In terms of *ZHT*, CEs/AEs present overall eastward/westward (positive/negative) heat transport in all seasons (Figure 12 a, b), and the maximum transport efficiency can be of the order of $10–20\times10^{12}$ W in the longitudes of 84°−88°E, corresponding to the eddy-rich regions in the northwestern bay (Figure 3). The seasonal *ZHT* of all eddies (Figure 12c) is less than $10\times10^{12}$ W. The mean *ZHT* (black line) of all eddies presents eastward transport in the western bay, whereas westward transport in the central and eastern parts of the bay. Owing to the seasonal variation of eddies in the Bay of Bengal, the seasonal *ZHT* of all eddies varies substantially. In autumn and winter, western and eastern parts of the bay correspond to eastward and westward heat transport, respectively, whereas the converse is true in spring, corresponding to westward and eastward heat transport. In addition, in summer, owing to the notable westward heat transport by AEs, the zonal heat transport of all eddies in the entire bay is almost negative.

Comparison with eddy-induced *ZHT* in the Bay of Bengal, eddy-induced *MHT* is substantially smaller (Figure 12 d-f). The magnitude of the seasonal *MHT* of CEs and AEs is almost below $5\times10^{12}$ W. CEs and AEs show almost opposite phase changes in the direction of *MHT*. In the northern bay to the north of 12°N, where most eddies move southward (Figure 5), the CEs and AEs exhibit northward (positive) and southward (negative) heat transport, respectively. Conversely, in the southern bay to the south of 12°N, where eddies tend to move northward, the CEs and AEs exhibit southward and northward heat transport, respectively. As the *MHT* of CEs is slightly larger than that of AEs, the *MHT* of all eddies is largely in the same direction as that of CEs (Figure 12f). The distribution characteristics of *MHT* are important for the heat balance within the entire bay. Northward heat transport can supplement the heat loss due to river runoff in the northern bay, while southward heat transport in the southern bay can balance the influx of high-temperature water carried by the inflow from regions outside of the bay. The seasonal *MHT* of all eddies also varies substantially. The magnitude of *MHT* in summer markedly exceeds that of other seasons. The maximum southward transport in the southern bay (about 8°−10°N) is nearly $10\times10^{12}$ W, which is related to the northward movement of the Sri Lanka cold eddy (Figure 10).



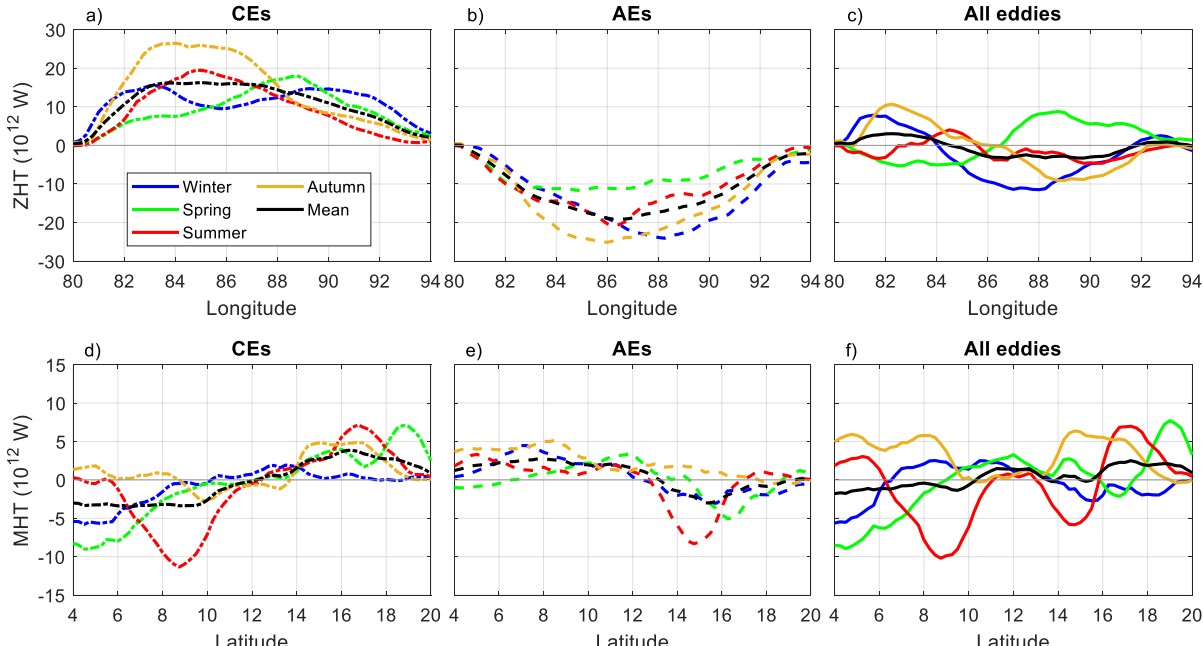

Figure 12: The meridionally integrated zonal heat transport (*ZHT*, upper panels) at different longitudes, and the zonally integrated meridional heat transport (*MHT*, bottom panels) at different latitutdes by cyclonic eddies (CEs), anticyclonic eddies (AEs), and all eddies in different seasons in the whole Bay of Bengal.

## 4.2 Eddy-induced salt transport and its seasonal variation

Salt transport $Q_s$ can be treated as an equivalent freshwater volume transport $F_w$ assuming conservation of mass across the transport section (Dong et al., 2014). The spatial distribution of eddy-induced freshwater volume transport $F_w$ in the Bay of Bengal is shown in Figure 13. In the part of the bay to the north of 12°N, the salinity anomalies caused by eddies are relatively monotonous with little interference by surface disturbances, and the freshwater transport $F_w$ is basically eastward/westward for CEs/AEs (CEs/AEs carry positive/negative salinity anomalies moving westward—westward/eastward salt transport— eastward /westward freshwater transport). The notable freshwater transport of all eddies is also concentrated in the northern part. In winter, AEs dominate the freshwater transport westward and southwestward; in spring, summer, and autumn, CEs donimate northeastward freshwater transport, which causes the salinity to decrease in the northern bay. The $F_w$ in the part of the bay to the south of 12°N is notably smaller than that in the northern part. The reason for the low freshwater transport in the southern part is related not only to the small number of eddies and their weak strength, but also to the complex structure of salinity anomalies caused by the eddies. In Section 3.2, we analyzed the spatial characteristics of the vertical salinity anomalies of eddies in the Bay of Bengal (Figure 10), and found that the salinity signals in the southern bay become turbulent owing to the invasion of the low-latitude equatorial circulation. Disturbance of salinity anomaly signals in the surface or subsurface waters reduces the salt transport capacity of CEs and AEs over the entire vertical structure.





Figure 13: Seasonal eddy-induced freshwater volume transport $F_w$ in the Bay of Bengal: (upper) results for cyclonic eddies, (middle) results for anticyclonic eddies, and (lower) results for all eddies. Here, $F_w = (F_{wm}, F_{wz})$ is a vector whose components are the meridional and zonal volume transports, the arrows indicate the transport direction, and the color indicates the transport magnitude.

Figure 14 shows the meridionally integrated zonal freshwater volume transport $ZWT = \int F_{wz} dy$ and the zonally

integrated meridional freshwater volume transport $MWT = \int F_{wm} dx$ caused by eddies within 4 subregions and across 5 sections, which represent the freshwater volume flux (unit: $m^3 \cdot s^{-1}$) in the entire meridional and zonal directions, respectively. The magnitude of the seasonal $ZWT$ of CEs is generally larger than that of AEs, thus the seasonal $ZWT$ of all eddies in the whole bay is eastward with high values (approximately $2.5 \times 10^3$ $m^3 \cdot s^{-1}$) in summer and autumn, consistent with CEs. The $ZWT$ of all eddies is only westward in the NB subregion in winter (dominated by AEs), the $ZWT$ in other subregions is

eastward. The eastward $ZWT$ is high in autumn and winter for both WB and CB, and in summer and autumn for SB. The seasonal $MWT$ of all eddies in the whole bay is small, except for the southward extreme of $1.6 \times 10^3$ $m^3 \cdot s^{-1}$ in winter. In the NB subregion, the $MWT$ of all eddies is southward (dominated by AEs) in winter and northward (dominated by CEs) in other seasons. In the WB, CB and SB subregions, the seasonal $MWT$ of all eddies is basically southward. The freshwater transport




across 5 cross-sections represents the eddy-induced freshwater exchange between the different subregions (lower panels in
Figure 14). The positive *MWT* across S1 section indicates that eddy carries freshwater from the WB region northward into
the NB region, the values are high in spring and summer. The *MWT* between NB and CB (across S2 section) is southward in
winter and spring, and northward in summer and autumn. The S3 and S4 sections both exhibit southward *MWT* with high
values in summer and autumn, meaning that eddy carries freshwater from the WB and CB regions southward into the SB
region. The *MWT* between WB and CB (across S5 section) is eastward (from WB to CB, dominated by CEs) with high
values in autumn and winter.

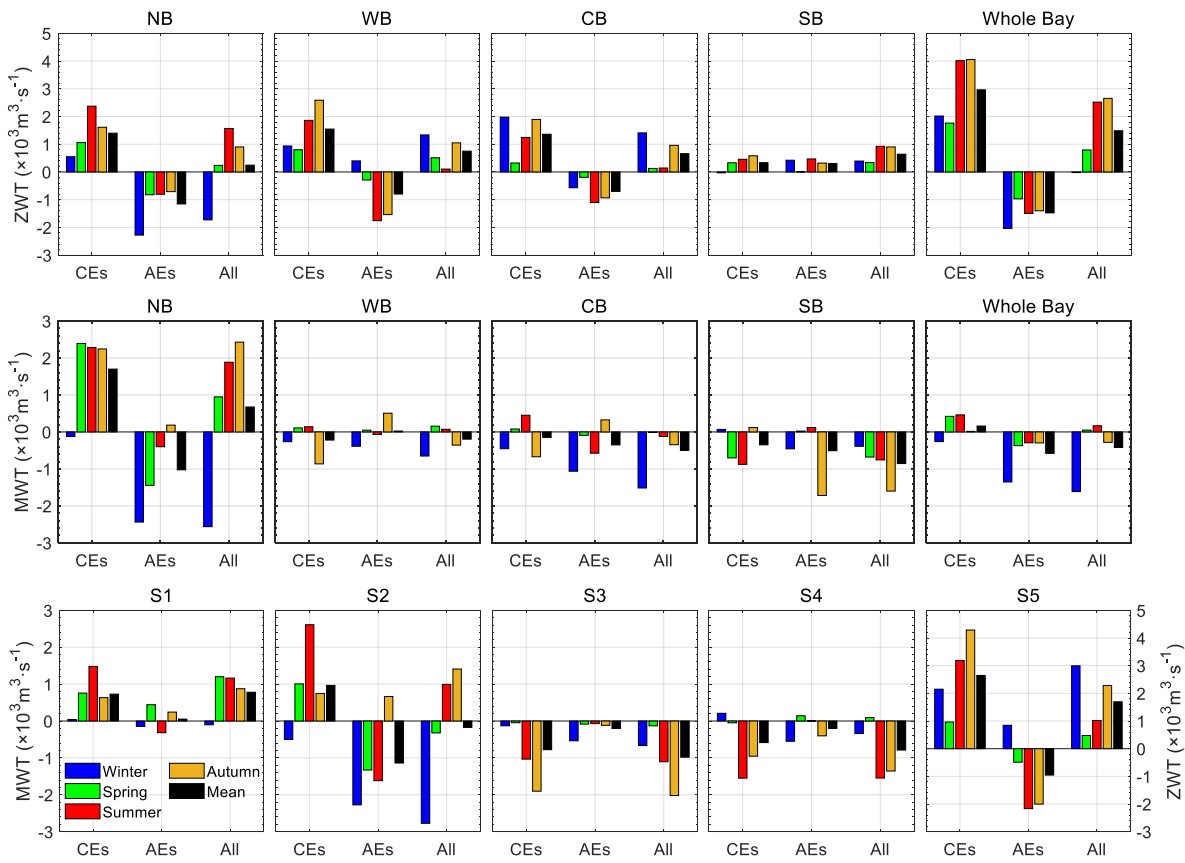

Figure 14: The meridionally integrated zonal freshwater volume transport (*ZWT*, upper) and zonally integrated meridional freshwater
volume transport (*MWT*, middle) by cyclonic eddies (CEs), anticyclonic eddies (AEs), and all eddies (All) in different seasons for 4
subregions and the whole Bay of Bengal. (lower) The seasonal freshwater volume transport by eddies across 5 sections. The four
subregions and 5 sections are shown in the lower right panel in Figure 10.

The seasonal eddy-induced *ZWT* at different longitudes and *MWT* at different latitutdes in the whole Bay of Bengal are
shown in Figure 15. In terms of *ZWT*, the overall transport capacity of CEs is greater than that of AEs. Therefore, the *ZWT*
direction of all eddies is largely consistent with that of CEs, e.g., except for the westward *ZWT* (negative value) in the eastern
part of the bay in winter, freshwater transport (positive value) is largely eastward throughout the entire bay in the rest of the
season. The *ZWT* associated with CEs in the central part of the bay is markedly greater than that in the western and eastern



parts. The maximum *ZWT* of CEs in autumn is greater than $10 \times 10^3$ m$^3 \cdot$s$^{-1}$ in the longitude of about 86°N. The *ZWT* of AEs is
625 relatively low, except in winter, the magnitude in other seasons is less than $3 \times 10^3$ m$^3 \cdot$s$^{-1}$. This is because the eastward
freshwater transport in the southern part of the bay weakens the westward transport in the entire meridional direction.

In terms of *MWT*, the magnitude of the mean transport is $<2 \times 10^3$ m$^3 \cdot$s$^{-1}$ for both CEs and AEs, which is substantially
smaller than that of *ZWT* (black lines in Figure 15). The *MWT* direction of CEs is largely the same as the direction of heat
transport, i.e., northward in the northern bay and southward in the southern bay. Southward freshwater transport of AEs is
630 presented almost in the entire bay. The combined effect of CEs and AEs result in northward *MWT* in the area to the north of
16°N, and southward *MWT* in the central and southern parts (Figure 15f). In addition, the *MWT* also shows obvious seasonal
variation. In winter, eddies cause greater southward transport that decreases from north to south. In spring, the northernmost
part of the bay has high northward transport, whereas the transport in the other areas is notably lower. The *MWT* in summer
and autumn is generally the same, i.e., northward in the northern bay and southward in the southern bay.

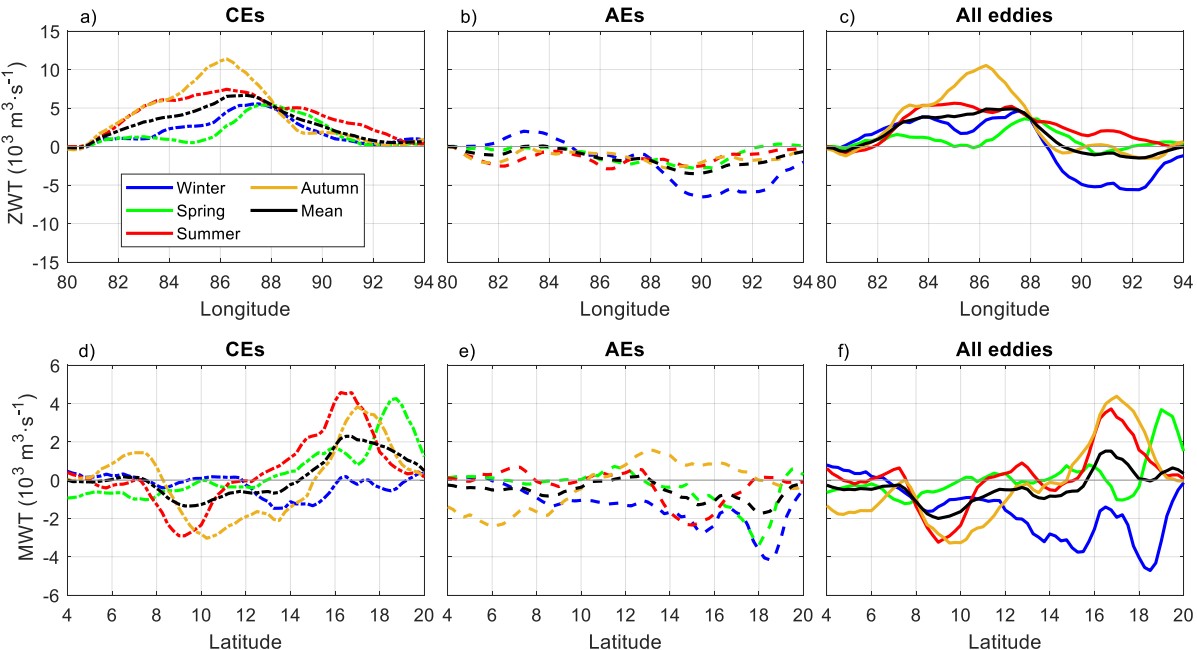

Figure 15: The meridionally integrated zonal freshwater volume transport (*ZWT*, upper panels) at different longitudes, and the zonally
integrated meridional freshwater volume transport (*MWT*, bottom panels) at different latitudes by cyclonic eddies (CEs), anticyclonic
eddies (AEs), and all eddies in different seasons in the Bay of Bengal.

Analysis of eddy-induced heat and salt transport and the seasonal variation showed that owing to obvious seasonal
changes of both the eddy activity and the vertical thermohaline structure in the Bay of Bengal, the heat and salt transport in
different seasons also changes substantially. The greatest heat transport is largely concentrated in the western and
northwestern parts of the bay, the seas to the east of Sri Lanka, and the region to the southeast outside of the bay where there
are either clustered high-intensity eddies or eddies with fast propagation speed that result in considerable heat transport. The
645 areas of greatest freshwater transport are largely concentrated in northern, northwestern, and eastern parts of the bay.
Conversely, the southern part of the bay show weak transport owing to the inconsistent salinity signal within eddies.



## 5 Summary and conclusions

The Bay of Bengal, occupying the eastern part of the tropical Indian Ocean, is characterized by the seasonal circulation and intense eddy activity throughout the year. Knowledge of the seasonal variation and vertical structure of eddies is vital both for comprehensive understanding of ocean dynamic processes and for analysis of the ocean circulation and energy transport in the Bay of Bengal. Based on satellite altimetry data in combination with Argo profile or 3D reprocessed thermohaline fields, the eddy synthesis method was used to construct vertical temperature and salinity structures of eddies in the Bay of Bengal, study their seasonal thermohaline variations as well as the heat and salt transport.

The mesoscale eddies in the Bay of Bengal were determined from satellite altimetry data spanning over 26 years from January 1993 to February 2019. The eddy result revealed that mesoscale eddy activity in the Bay of Bengal has evident seasonal variation, which has notable impact on the circulation system within the entire bay. Eddy intensities in spring and summer is relatively strong, and that in autumn and winter is relatively weak. During the winter Northeast Monsoon, the bay is dominated by a clear cyclonic gyre, accompanied by abundant CEs in the western bay; while a persistent AE form in the northern bay. In spring premonsoon, a basin-scale anticyclonic gyre dominates the bay, accompanied by many strong AEs in the northwestern bay; while CEs are prone to appear in the northernmost part of the bay. During the summer Southwest Monsoon, the overall circulation structure is controlled by two north–south CEs and a strong AE in the middle. Owing to the strong Southwest Monsoon is obstructed by the land area of Sri Lanka, a presistant and strong CE (Sri Lanka cold eddy) with high eddy kinetic energy often appears to the east of Sri Lanka. During the autumn postmonsoon, the Sri Lanka cold eddy moves northward and interacts with the reversed EICC before finally disappearing in the western parts of the bay. Generally, there are three main areas of distribution of mesoscale eddies in the Bay of Bengal. One is the EICC region in the west and northwest of the bay, indicating that variation or reversal of the EICC will often shed rich eddy structures. Another region is the northeastern part of the bay, which is affected by the instability of the eastern flow in the bay or by the intrusion of the river discharge in the northeast of the bay. The third region is seas to the east of Sri Lanka, where there are strongs CEs in summer and AEs in autumn. The eddy propagation speed in different seasons also shows obvious differences. The westward speed of eddies is fastest in winter, followed by spring and autumn, and slowest in summer. The meridional speed of eddies is faster in winter and spring, and eddies move southward outside the bay in summer and autumn.

An investigation of vertical thermohaline structure of the eddies shows that owing to the seasonal eddy activities, the vertical thermohaline structure of eddies in the Bay of Bengal has obvious seasonal variation, as well as some differences for the northern and southern subregions. The extrema of temperature anomalies $\theta'$ caused by eddies are located at shallower depth (approximately 80/100 dbar for CEs/AEs) in the southern bay than that (approximately 100/120 dbar for CEs/AEs) in the northern bay, due to the shallower thermocline in the southern bay. For eddies in the northern bay, the $\theta'$ of CEs and AEs are both maximum in spring, up to $\pm 2.5°C$, and minimum in winter, about $\pm 1.2°C$, and about $\pm 2°C$ in summer and autumn, respectively. For eddies in the southern bay, the $\theta'$ of CEs is the largest in summer, reaching $-3°C$, the smallest in winter, less than $-2°C$; the $\theta'$ of AEs is larger in summer and autumn, around $+2°C$, and smaller in winter and spring, around $+1.5°C$. The salinity anomalies $S'$ of eddies in the northern and southern subregions present larger differences. CEs (AEs) produce notable positive (negative) signals at the subsurface in the northern bay. The maximum $S'$ of CEs in spring can exceed $+0.3$ psu, whereas it is around $+0.25$ psu in autumn, and weakest in winter is only $+0.2$ psu; the extremum of negative $S'$ in AEs in autumn can reach $-0.35$ psu, values are around $-0.25$ psu in spring and summer, and $-0.2$ psu in winter. For $S'$ in the



southern bay, the magnitude of $S'$ signals is significantly small for both CEs and AEs due to the small vertical gradient of salinity in the southern bay.

By combining the temperature and salinity anomalies of each eddy, provided by the weekly ARMOR3D thermohaline field data, with the details of eddy movement (propagation trajectory), provided by gridded multimission altimeter products, we estimated the eddy-induced heat and salt transport in different areas of the Bay of Bengal. CEs/AEs generally present eastward/westward heat transport because they carry negative/positive heat anomalies westward across the bay. The greatest heat transport is largely concentrated in the western and northwestern parts of the bay, the seas to the east of Sri Lanka, and the region to the southeast outside of the bay where there are either clustered high-intensity eddies or eddies with fast propagation speed that result in considerable heat transport. Owing to seasonal variations of both the eddy activity and the vertical thermohaline structure in the Bay of Bengal, the heat transport in different seasons also changes substantially. The magnitude of the seasonal $ZHT$ of CEs and AEs in the whole bay is in the order of $10 \times 10^{12}$ W, with higher values in autumn and winter and smaller values in spring and summer. Because the $ZHT$ directions of CEs and AEs are opposite, the $ZHT$ of all eddies in the whole bay is small, and the value in each season is below $2 \times 10^{12}$ W. The magnitude of the seasonal $MHT$ of all eddies in the whole bay is also small, mostly below $1 \times 10^{12}$ W. However, the NB and SB subregions present high $MHT$ with obviously seasonal variation.

Eddy-induced salt transport is treated as an equivalent freshwater volume transport in this study. The areas of greatest freshwater transport are largely concentrated in the northern, northwestern, and eastern parts of the bay. Conversely, the southern part of the bay show weak transport owing to the inconsistent salinity signal within eddies. The magnitude of the seasonal $ZWT$ of CEs is generally larger than that of AEs, thus the seasonal $ZWT$ of all eddies in the whole bay is eastward with high values (approximately $2.5 \times 10^3$ m$^3 \cdot$s$^{-1}$) in summer and autumn, consistent with CEs. The $ZWT$ of all eddies is only westward in the NB subregion in winter (dominated by AEs), the $ZWT$ in other subregions is eastward. The seasonal $MWT$ of all eddies in the whole bay is small, except for the southward extreme of $1.6 \times 10^3$ m$^3 \cdot$s$^{-1}$ in winter. These results of eddy-induced heat-salt transport in the Bay of Bengal based on measured temperature and salt data are different to the results obtained through theoretical calculation by Gonaduwage et al. (2019). The findings show that diverse seasonal changes of temperature and salinity in the Bay of Bengal might cause substantial deviation in eddy-induced heat/salt transport estimated theoretically. This work provides data that could support further research on the heat and salt balance of the entire Bay of Bengal.

## Acknowledgments

This study is supported by the National Natural Science Foundation of China (Grant No. 42106178), the Basic Scientific Fund for National Public Research Institutes of China (Grant No. 2020Q07), and the Shandong Provincial Natural Science Foundation (Grant No. ZR2021QD006). We acknowledge all the data providers for the data utilized in this study.

## Code/Data availability

The altimeter products (SEALEVEL_GLO_PHY_L4_REP_OBSERVATION_008_47) and the Global ARMOR3D L4 Reprocessed dataset (MULTIOBS_GLO_PHY_REP_015_002) used here are distributed by the European Copernicus



Marine Environment Monitoring Service (CMEMS, http://marine.copernicus.eu). The Argo profiles are provided by Coriolis
Global Data Acquisition Center (http://www.coriolis.eu.org ).

## Author contribution

JY and JZ designed the experiments and WC carried them out. WC performed the data analyses and drafted the paper.
WC prepared the manuscript with contributions from all co-authors.

## Competing interests

The authors declare that they have no conflict of interest.

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
