# Peer review of "Seasonal variation of eddy activity and associated heat/salt transport in the Bay of Bengal based on satellite, Argo and 3D reprocessed data"

_EGUsphere, 2022_

## Author Comment (AC1)

Responds to the reviewer's comments:

**To Reviewer #1:**

Based on the reviewers' comments, the main revisions of the new manuscript are as follows:

(1) The redundant description of the manuscript is compressed, especially for the Section 3 (such as seasonal spatial distribution of eddies and seasonal variation of vertical structure are simplified, and Figure 4 and 6 in the old manuscript are removed). Furthermore, the new results and findings are highlighted in the manuscript, such as the seasonal variations of eddy movements (Section 3.1), the seasonal thermohaline properties in north and south bay and their regional variations (Section 3.2).

(2) We have removed the analysis of eddy-induced heat/salt transport in different sub-regions (Figure 11 and 14 in the old manuscript) in Section 4, by introducing the divergence of heat/salt transport to estimate the impact of eddy movements. New additions are marked in green in Section 4.

(3) Some details about 3D eddy reconstruction have been added in Section 2.2.2.

(4) In order to be consistent with other studies (Gonaduwage et al., 2019; Gulakaram et al., 2020; George et al., 2019, JPO), eddy-induced salt transport (Section 4.2) is represented by salt transport $Q_s$ itself (Figure 12 and 13), discarding the freshwater transport $F_w$. In the analysis of the divergence of eddy salt transport, the equivalent freshwater flux is used for comparison with net freshwater flux at surface (Figure 11).

Below are the point-to-point responses.

**Response to comment #1:** *In its present form, this manuscript consists primarily of a long (over 400 lines) results section (Sections 3 and 4). I found parts of this quite difficult to follow and fairly repetitive……*

The redundant description of the manuscript is compressed, especially for the Section 3. Furthermore, the new results and findings are highlighted in the manuscript, such as the seasonal variations of eddy movements (Section 3.1), the seasonal thermohaline properties in north and south bay and their regional variations (Section 3.2).

The basic logic of this research is to combine seasonal eddy activities (Section 3.1) and seasonal vertical thermohaline structures (Section 3.2) within eddies to estimate seasonal eddy-induced heat/salt transport (Section 4) in the Bay of Bengal.

Section 4 presents the spatial distribution of eddy-induced heat/salt transport, the zonal and meridional heat/salt transports (e.g., *ZHT*, *ZST*, *MHT*, *MST*), and the divergence of eddy heat/salt transport. We have removed the analysis of eddy-induced heat/salt transport in different sub-regions, by introducing the divergence of heat/salt transport to compare it with the Air-Sea heat flux and net freshwater flux, the impact of eddy movements is estimated. New additions are marked in green in Section 4.

In addition, at the end of the manuscript (Section 5 Summary and Discussion), we have added a discussion of the results.

**Response to comment #2:** *The authors quantify transport in Section 4 – are these big numbers? Would we expect these transports to make a big difference to the temperature and salinity distributions in the BoB? ……*

To estimate the impact of heat/salt transports by eddy movements in the Bay of Bengal, the divergence of eddy heat/freshwater transports were calcualted. The 10−20 W·m$^{-2}$ value of the eddy-induced heat flux is comparable in magnitude with the annual mean Air-Sea net heat flux, implying that the mesoscale eddies can exert a strong impact on the oceanic heat transport and redistribution in the Bay of Bengal. Notable, the high eddy-induced ocean heat gain in the eastern seas of Sri Lanka in summer suggests that eddy activities would somewhat balance the heat loss due to the intrusion of cold water carried by the Southwest Monsoon Current. In addition, the magnitude of freshwater gains and losses in the southern part of the bay is small in all seasons, which is mainly related to the weaker salt transport caused by the inconsistency of salinity signal within eddies. However, compared with the north-south variation of the annual mean net freshwater flux at surface, the spatial distribution of eddy-induced freshwater flux (the magnitude is generally $0-20\times10^{-6}$ kg·m$^{-2}$·s$^{-1}$, seasonal variation is higher, up to $50\times10^{-6}$ kg·m$^{-2}$·s$^{-1}$ regionally) shows an east-west variation, which indicates that mesoscale eddies plays an important role in maintaining the east-west freshwater or salt balance in the Bay of Bengal.

The new contents are marked in green in Section 4.

**Response to comment #3:** *I think that the method needs a little more explanation and justification in places. Section 2.2.2 in particular seemed to be a brief explanation of a complex process……*

Some details have been added in Section 2.2.2.

In fact, mesoscale eddies capture only few Argo profiles in a single day (Figure 2). With one or few Argo profiles, we cannot obtain the vertical thermohaline structure of one eddy (only a profile data inside the eddy). In order to study the 3D thermohaline structure of eddies in a certain region, the basic assumption is that all eddies exhibit similar 3D structures, so that all Argo vertical profiles that fall inside CEs and AEs are interpolated to create an average CE and an average AE 3D profile. Therefore, by matching the identified eddies with Argo profiles in a long time, a large number of Argo profiles within eddies can be obtained. Argo profiles captured by eddies are scattered (spatially nonuniform), it is necessary to transform these Argo profiles into a unified eddy-center coordinate, so as to combine the vertical temperature and salt information provided by all profiles to obtain the average 3D structures of eddies. Specifically, for each Argo profile matched by an eddy, we calculated the relative zonal and meridional distances ($\Delta$x, $\Delta$y) to the eddy center. The relative distances were normalized relative to the eddy radius (nondimensionalization). Thus, all profiles were transformed into the normalized eddy coordinate space (norm $\Delta$x, norm $\Delta$y), as shown below.

[Figure]

The relative positions of all profiles the normalized eddy coordinate space for the 3D eddy reconstruction of the cyclonic eddy (left panel) and the anticyclonic eddy (right panel), the color indicates the relative distance (dimensionless). The results for Argo profiles with a distance of 1.5 radii from the eddy core are given here, in fact we only selected profiles within eddies for calculating thermohaline anomalies of composite eddies (Section 3.2 in new manuscript).

Then, for each depth layer (from the surface to 1000 dbar with an interval of 10 dbar), and $\theta$, $S$ and $\theta'$, $S'$ data of Argo profiles were mapped onto 0.1×0.1 grid using inverse distance weighting interpolation. Finally, all the depth layers with uniform thermohaline anomaly fields were combined to obtain the 3D eddy structure. Since the Argo float only provide one-dimensional information on the profile, and Argo profiles are scattered, we can only reconstruct one 3D thermohaline structure of eddies in a region by the above method. Considering the hydrological differences from north of the bay to the south, here the Bay of Bengal is divided into north and south subregions with 12°N as the boundary to study the eddy 3D structure of each subregion. In order to highlight the difference between the north and the south, this study only gives the mean vertical profiles of the temperature/salt anomaly but does not give the results of the vertical section (such as Figure 8 and 9 in Gulakaram et al. (2020) and Figure 8 in Lin et al. (2019)).

The ocean reprocessed data provide the 3D temperature and salinity field data covering the entire space. This allows us to obtain the 3D thermohaline structure of the surface eddies captured by the satellite altimetry by matching the eddy results with the reprocessed 3D field data. We matched the eddy results identified from daily SLA fields with the weekly 3D field data at the closest time such that we could obtain the 3D temperature and salt structure of each eddy (Figure 2a). Similar to the handling of Argo profiles, all eddies were classified by season, and the 3D structures of all vortices in a season were averaged and used for comparison with the reconstruction results of Argo profiles.

**Response to comment #4:** *Finally, I encourage the authors to provide some more justification of their decision to divide the BoB up into five sub-regions, as outlined in Figure 10……*

We have removed the analysis of eddy-induced heat/salt transport in different sub-regions in Section 4. The *ZHT*/*MHT* in Figure 10 and *ZST*/*MST* in Figure 13 show the eddy-induced heat/salt transport through a certain section (in the entire meridional or zonal directions). To estimate the impact of heat/salt transports by eddy movements in the Bay of Bengal, the divergence of eddy heat/freshwater transports (Figure 11) were calcualted and compared with the Air-Sea heat flux and net freshwater flux. Based on this analysis, the impact of heat/salt transports by eddy movements on the local area can be observed more clearly.

**Response to comment #5:** *Line 122. Hydrography should be presented as conservative temperature and absolute salinity.*

In order to be consistent with most current eddy studies (to facilitate comparison of results), the potential temperature and practical salinity (psu) are still used in this paper. And by calculating conservative temperature and absolute salinity, they have very little difference on the results.

**Response to comment #6:** *Figures 2, 4 and 9. I recommend plotting these figures using a diverging colour bar (e.g. red to blue) with zero in the middle.*

These figures have been replotted using a BOD color bar, in order to distinguish them from the divergence of eddy-induced heat/salt transport (Figure 11, the diverging color bar is used).

**Response to comment #7:** *Line 240. Do the authors mean significantly in the statistical sense? Or do they just mean "a lot"?*

It means that eddy amplitude varies greatly in different season. We have revised the wording "vary greatly" to avoid ambiguity.

**Response to comment #8:** *Line 243. Throughout the manuscript, the authors appear to use eddy intensity and eddy amplitude interchangeably. I would recommend sticking to amplitude to improve clarity.*

"Eddy intensity" has been modified to "eddy amplitude" throughout the manuscript.

**Response to comment #9:** *Line 250. I wonder whether Table 1 wouldn't work better as a figure? Maybe a series of bar charts?*

Table 1 has been replaced by a series of bar charts (Figure 3) in the new manuscript.

**Response to comment #10:** *Figure 4. The arrows on this figure don't show up very well against the coloured background.*

Figure 4 in the old manuscript has been moved to the Supplementary Materials (Figure S3). The BOD color bar is used for the background field (SLA) to better display the flow field (arrows).

**Response to comment #11:** *Line 309. What do the authors mean by a "relatively concentrated distribution"?*

It means many AEs are clustered in the eastern seas of Sri Lanka. In the new manuscript we have simplified this part of the description.

**Response to comment #12:** *Line 361. This paragraph promises discussion of how the BoB's water masses influence eddy properties, but no such discussion ever appears. Instead, this paragraph is mainly introduction-style material about water mass hydrography.*

To highlight the new results and findings of this paper, this introductory text and Figure 6 in the old manuscript have been removed.

**Response to comment #13:** *Figures 7 and 8. The shading representing standard deviation is too faint to see clearly.*

The two figures have been redrawn as Figures 6 and 7 in the new manuscript.

**Response to comment #14:** *Line 420. What do the authors mean by a "concentrated" eddy?*

It means clustered eddies. Here is a comparison with the Sri Lanka Dome in summer (many CEs are clustered there) to illustrate that the other seasons lack clustered and strong eddies, so their salinity signals are weaker. We have revised inappropriate expressions here.

**Response to comment #15:** *Line 423. "Difficult to cause a large salinity change" doesn't make sense.*

This has been removed.

**Response to comment #16:** *Line 455. I don't quite understand what the authors mean by "confused positive salinity anomalies".*

For AEs, in contrast to the negative salinity anomalies in the northern part of the bay, some disordered positive salinity anomalies are present in the southern part. The expression here is modified to "disordered positive salinity anomalies".

**Response to comment #17:** *Line 483. What do the authors mean by "almost present"?*

The text has been modified to "CEs/AEs present eastward/westward heat transport in most regions".

**Response to comment #18:** *Line 496. What do the authors mean by "it" in this sentence?*

It means "the eastern bay", the eastern bay generally corresponds to westward heat transport in autumn and winter, due to the prevalence of AEs moving westward in the seasons (Cheng et al., 2018).
The text has been revised.

**Response to comment #19:** *Line 573. I don't know what "monotonous" means here.*

It means "uniform", the word has been replaced by "uniform".

**Response to comment #20:** *Line 603. The word "basically" sounds vague here and should probably be avoided.*

This part of the content was deleted.

**Response to comment #21:** *Line 706. The comparison with the results of Gonaduwage et al (2019) would make an interesting discussion point. It would be nice to see this expanded.*

Gonaduwage et al (2019) adopted Stammer's (1998) eddy diffusivity method for the basin-scale calculations and look at the eddy time scales in the Bay of Bengal using altimetry data. They did not identify the mesoscale eddies in the Bay of Bengal, nor did they analyze the structure of temperature and salinity anomalies within the eddies. On the basis of eddy diffusivity derived "mixing length" arguments, they estimated eddy-induced heat and salt transport in the Bay of Bengal based on integral time scales of sea surface variability and near surface eddy kinetic energy. Actually, their result is a kind of theoretical analysis result. In their assumption, the direction of eddy transport depends only on the gradient of the background temperature and salt field (T and S, Equation 1-4 in their paper). In fact, different types of eddies (e.g., CEs or AEs, they carry different temperature and salinity waters) move in different directions, and the directions for the transport are completely different.

In our research, eddies were identified from daily SLA fields and eddy propagation trajectories were tracked in the continuous time series. Furthermore, through matching identified eddies with Argo profile or 3D reprocessed thermohaline fields, the eddy synthesis method was used to construct vertical temperature and salinity structures of eddies. At last, by combining the temperature and salinity anomalies of eddies with the details of eddy movement (propagation trajectory), we estimated the eddy-induced heat and salt transport in different areas of the Bay of Bengal. Our work can be viewed as a direct estimation of eddy transport based on observational data. Due to different methods used for eddy-induced transport, our results differ somewhat from Gonaduwage et al (2019). Although there are some differences in transport directions, both the two results show that the high transport is distributed in eddy-rich regions, e.g., the western, northern parts of the bay, the seas to the east of Sri Lanka, and the region to the southeast outside of the bay.

The difference of transport direction between the two results obtained from theoretical analysis and observational data may be mainly caused by two reasons. One is that Gonaduwage et al (2019) did not specifically separate mesoscale eddies from the background field (simply using eddy kinetic energy $K_E$ to represent eddy activities) and not consider the direction of eddy movements. The difference in the direction of eddy movements will bring about the essential difference in the direction of heat and salt transport. The theoretical analysis may not be able to accurately obtain the direction of eddy movements, especially in the coastal waters. The second is that there may be a deviation in the estimation of the temperature and salinity anomalies caused by eddies in the theoretical analysis (they only considered the gradient of mean temperature and salinity), especially the temperature and salinity anomaly signals in different regions are significantly different. This will affect the result of the direction and magnitude of the eddy-induced transport. Therefore, in our research, we analyzed the temperature and salinity anomalies of eddies and the details of eddy movement (propagation trajectory) in different regions. Based on these analyses, eddy-induced heat and salt transport was estimated. Compared to theoretical estimation, our research provides a more direct computational result based on observational data. The findings also show that diverse seasonal changes of temperature and salinity in the Bay of Bengal might cause substantial deviation in eddy-induced heat/salt transport estimated theoretically.

A brief discussion about the two results is added in Section 5.

Other minor grammar and expression errors modified in the paper are not listed here.

Special thanks to you for your good comments.

---

## Author Comment (AC2)

Responds to the reviewer's comments:

**To Reviewer #2:**

Based on the reviewers' comments, the main revisions of the new manuscript are as follows:

(1) The redundant description of the manuscript is compressed, especially for the Section 3 (such as seasonal spatial distribution of eddies and seasonal variation of vertical structure are simplified, and Figure 4 and 6 in the old manuscript are removed). Furthermore, the new results and findings are highlighted in the manuscript, such as the seasonal variations of eddy movements (Section 3.1), the seasonal thermohaline properties in north and south bay and their regional variations (Section 3.2).

(2) We have removed the analysis of eddy-induced heat/salt transport in different sub-regions (Figure 11 and 14 in the old manuscript) in Section 4, by introducing the divergence of heat/salt transport to estimate the impact of eddy movements. New additions are marked in green in Section 4.

(3) Some details about 3D eddy reconstruction have been added in Section 2.2.2.

(4) In order to be consistent with other studies (Gonaduwage et al., 2019; Gulakaram et al., 2020; George et al., 2019, JPO), eddy-induced salt transport (Section 4.2) is represented by salt transport $Q_s$ itself (Figure 12 and 13), discarding the freshwater transport $F_w$. In the analysis of the divergence of eddy salt transport, the equivalent freshwater flux is used for comparison with net freshwater flux at surface (Figure 11).

Below are the point-to-point responses.

1    **Response to comment #1:** *The results presented are too lengthy and often over-descriptive. Several features of the eddy structure and variability in the Bay of Bengal have been presented is several papers by Cheng et al. and Cui et al. The new results that have emerged consequent to the separation of the analysis into a seasonal cycle need to be highlighted and repletion be avoided.*

The redundant description of the manuscript is compressed, especially for the Section 3. Furthermore, the new results and findings are highlighted in the manuscript, such as the seasonal variations of eddy movements (Section 3.1), the seasonal thermohaline properties in north and south bay and their regional variations (Section 3.2).

The basic logic of this research is to combine seasonal eddy activities (Section 3.1) and seasonal vertical thermohaline structures (Section 3.2) within eddies to estimate seasonal eddy-induced heat/salt transport (Section 4) in the Bay of Bengal.

Section 4 presents the spatial distribution of eddy-induced heat/salt transport, the zonal and meridional heat/salt transports (e.g., *ZHT*, *ZST*, *MHT*, *MST*), and the divergence of eddy heat/salt transport. We have removed the analysis of eddy-induced heat/salt transport in different sub-regions, by introducing the divergence of heat/salt transport to compare it with the Air-Sea heat flux and net freshwater flux, the impact of eddy movements is estimated. New additions are marked in green in Section 4.

2    **Response to comment #2:** *Significant new contribution from this study is the estimation of heat and salt transports. These transports, however, appear as patches of relatively short spatial extent. What are implications of these transport? Do they affect the SST distribution? Do they affect the heat budget of the Bay of Bengal? The authors may also consider separating the heat and salt fluxes into individual contributions due to cyclonic and anticyclonic eddies.*

To estimate the impact of heat/salt transports by eddy movements in the Bay of Bengal, the divergence of eddy

heat/freshwater transports were calcualted. The 10−20 W·m$^{-2}$ value of the eddy-induced heat flux is comparable in magnitude with the annual mean Air-Sea net heat flux, implying that the mesoscale eddies can exert a strong impact on the oceanic heat transport and redistribution in the Bay of Bengal. Notable, the high eddy-induced ocean heat gain in the eastern seas of Sri Lanka in summer suggests that eddy activities would somewhat balance the heat loss due to the intrusion of cold water carried by the Southwest Monsoon Current. In addition, compared with the north-south variation of the annual mean net freshwater flux at surface, the spatial distribution of eddy-induced freshwater flux (the magnitude is generally 0−20×10$^{-6}$ kg·m$^{-2}$·s$^{-1}$, seasonal variation is higher, up to 50×10$^{-6}$ kg·m$^{-2}$·s$^{-1}$ regionally) shows an east-west variation, which indicates that mesoscale eddies plays an important role in maintaining the east-west freshwater or salt balance in the Bay of Bengal.

In the analysis of the spatial distribution of eddy-induced heat/salt transport (Figure 9 and 12), the zonal and meridional heat/salt transports (e.g., *ZHT, ZST, MHT, MST*, Figure 10 and 13), we given the individual contributions due to CEs and AEs separately.

The new contents are marked in green in Section 4.

3     **Response to comment #3:** *'Sri Lanka eddy' described in this paper is well known as Sri Lanka Dome (Vinayachandran and Yamagata, JPO, 1998). See the recent paper by Cullen and Shroyer (2022) for additional references. It is suggested that the terminology that is in practice be used.*

"Sri Lanka eddy" has been replaced by "Sri Lanka Dome" in the new manuscript, and the references have also been added.

4     **Response to comment #4:** *Justify why the removal of seasonal cycle give the mesoscale structure. Past studies have removed 3 or more harmonic or applied appropriate filters to extract mesoscale variations.*

Because in Argo and reanalysis data, the thermohaline data contains obviously seasonal variations, especially for the upper ocean. Therefore, in order to obtain the temperature and salt changes caused by mesoscale eddies, these large-scale seasonal changes should be removed from the temperature and salt data. Otherwise, the seasonal signal will be included in the eddy-induced temperature and salinity anomalies.

5     **Response to comment #5:** *What are stable eddies?*

Here, stable eddies are relative to small-scale turbulent signals. Because the messy eddy trajectories obscure the distribution characteristics of eddy activities (Figure S1). In order to understand the seasonal distribution characteristics of eddies in the Bay of Bengal more intuitively, we used monthly averaged SLA fields to identify eddies that occur frequently in certain regions (here we call them "the monthly eddies").

The stable eddies have been removed to avoid ambiguity.

6     **Response to comment #6:** *Is this a schematic? Or is this data for a selected day? Description of line legends A to F is missing.*

Figure 2 is a case of matching identified eddies with Argo profiles and 3D thermohaline field on 20$^{th}$ May 2017. Corresponding date have been added to the text. The description of legend A to F has also been added to the text.

7     **Response to comment #7:** *This sentence is not clearly written. Replace 'choosed' with 'chosen'.*

Modified as suggested.

8   **Response to comment #8:** *The primed quantities have not been defined appropriately.*

We have defined all primed quantities in Equations 1 and 2.

9   **Response to comment #9:** *Please replace 'changeable' with variable and correct the sentence.*

Modified as suggested.

10  **Response to comment #10:** *Replace Sri Lanka cold eddy to Sri Lanka Dome.*

Modified as suggested.

11  **Response to comment #11:** *"However, eddies will exchange heat and salt …". This is not apparent. Please justify with appropriate references.*

We have simplified the text at the beginning of Section 4.

Other minor grammar and expression errors modified in the paper are not listed here.
Special thanks to you for your good comments.

---

## Author Response (AR2)

Dear Editor:

Thank you for your letter and for the reviewer's comments concerning our manuscript ID **egusphere-2022-453** entitled "Seasonal variation of eddy activity and associated heat/salt transport in the Bay of Bengal based on satellite, Argo and 3D reprocessed data". The main corrections in the paper and the responds to the reviewer's comments are as following:

Responds to the reviewer's comments:

**Minor comment:** *Section 2.2.2, I understand now that they authors are here creating a single composite cyclonic and anti-cyclonic eddy from the Argo observations. I would ask them to just make this point explicitly at the very beginning of the section: this would make the following method even clearer.*

Combined with the satellite altimetry data and the Argo profiles, composite 3D structures of a single CE and a single AE were created based on eddy synthesis method in the study, respectively. The 3D structures of eddies were constructed by surfacing the Argo float profiles into SLA-based eddy areas, as shown in Figure 2. We considered the detection results (from daily SLA fields) of the long-lived eddy (eddy trajectories with lifetime ≥30 days) to match the Argo profiles on the same day, and selected Argo profiles with a distance of <1.5 radii from eddy center for vertical eddy structure analysis. By matching identified eddies with Argo profiles in a long time (from the year 2001 to 2019), a large number of Argo profiles within eddies can be obtained. Consequently, 3882 and 4097 Argo profiles were selected for cyclonic and anticyclonic eddy reconstruction, respectively. These Argo profiles were interpolated to create an average CE and an average AE 3D profile.

The explanatory sentence has been added and marked in green at the beginning of the Section 2.2.2.

Special thanks to you for your valuable comment, it is very helpful for revising and improving our paper.

We appreciate for Editors/Reviewers' warm work earnestly, and hope that the correction will meet with approval. Please feel free to contact us with any questions and we are looking forward to your consideration. Once again, thank you very much for your comments and suggestions.

Sincerely,
Wei Cui